# Surface tension-assisted additive manufacturing

Héloïse Ragelle[1,2], Mark W. Tibbitt [1,7], Shang-Yun Wu[1], Michael A. Castillo[1], George Z. Cheng[3], Sidharta P. Gangadharan[4], Daniel G. Anderson[1,2,5], Michael J. Cima[1,6] & Robert Langer[1,5]

The proliferation of computer-aided design and additive manufacturing enables on-demand fabrication of complex, three-dimensional structures. However, combining the versatility of cell-laden hydrogels within the 3D printing process remains a challenge. Herein, we describe a facile and versatile method that integrates polymer networks (including hydrogels) with 3D-printed mechanical supports to fabricate multicomponent (bio)materials. The approach exploits surface tension to coat fenestrated surfaces with suspended liquid films that can be transformed into solid films. The operating parameters for the process are determined using a physical model, and complex geometric structures are successfully fabricated. We engineer, by tailoring the window geometry, scaffolds with anisotropic mechanical properties that compress longitudinally (~30% strain) without damaging the hydrogel coating. Finally, the process is amenable to high cell density encapsulation and co-culture. Viability (>95%) was maintained 28 days after encapsulation. This general approach can generate biocompatible, macroscale devices with structural integrity and anisotropic mechanical properties.

[1] The David H. Koch Institute for Integrative Cancer Research, Massachusetts Institute of Technology, 500 Main St Cambridge, Cambridge, MA 02142, USA. [2] Department of Anesthesiology, Boston Children's Hospital, Harvard Medical School, 300 Longwood Ave Boston, Boston, MA 02115, USA. [3] Department of Medicine, Pulmonary, Allergy, and Critical Care Medicine, Duke University School of Medicine, 20 Duke Medicine Circle Durham, Durham, NC 27710, USA. [4] Department of Surgery, Beth Israel Deaconess Medical Center, Harvard Medical School, 330 Brookline Ave Boston, Boston, MA 02215, USA. [5] Department of Chemical Engineering, Massachusetts Institute of Technology, 500 Main St Cambridge, Cambridge 02142 MA, USA. [6] Department of Materials Science and Engineering, Massachusetts Institute of Technology, 500 Main St Cambridge, Cambridge, MA 02142, USA. [7] Present address: Macromolecular Engineering Laboratory, Department of Mechanical and Process Engineering, ETH Zürich, Sonneggstrasse 3, 8092 Zürich, Switzerland. These authors contributed equally: Héloïse Ragelle, Mark W. Tibbitt. Correspondence and requests for materials should be addressed to R.L. (email: rlanger@mit.edu)

An ideal engineered biomaterial for organ repair or tissue replacement should incorporate a scaffolding structure that recapitulates organ geometry, physical properties (such as toughness or elasticity), and possible mechanical anisotropy, as well as a tailored microenvironment for cells to encourage biologic function and integration with the host[1]. The emergence of additive manufacturing, i.e., three-dimensional (3D) printing, has enabled the design of (bio)materials with high precision, fine control of architecture, and mechanical properties, as well as patient-specific geometry informed by advanced bio-medical imaging[2–4]. In parallel, natural and synthetic hydrogels mimicking critical aspects of the extracellular matrix, have been employed broadly in regenerative medicine as they integrate well in vivo and can be engineered with a range of elastic moduli and surface chemistries specific to the cell type of interest[5]. Integrating the full utility of cell-laden hydrogels within the 3D printing process remains, however, an active challenge. Realization of the full 3D structure of living tissue with relevant mechanical properties is particularly difficult.

3D bioprinting has emerged as a means to fabricate geometrically defined and cell-laden biomaterials. Predominant strategies include inkjet bioprinting, microextrusion and laser-assisted bioprinting, and rely on layer-by-layer deposition of hydrogels and cells[6,7]. Despite significant progress, the hydrogel-based 3D-printed constructs commonly possess insufficient mechanical stability and structural integrity, especially when printing with low modulus materials ($E < 100$ kPa)[6,8]. In addition, as the printing is done on a planar surface, the possibility of material shapes and designs is limited and complex mechanical structures are difficult to print. Embedded bioprinting approaches have been developed wherein material is printed directly into a temporary support bath. Examples include a shear-thinning hydrogel or a thermosensitive gelatin microparticle suspension that provides a temporary mechanical support during printing and is removed after crosslinking, enabling omni-directional printing[8,9]. This technique has enabled the fabrication of elegant and complex soft materials. In other work, polylactic acid or hydrogel-based scaffolds were printed with micron-scale pores and immersed in a coating solution post-fabrication. The coatings improved cell adhesion and bio-integration of the 3D-printed structures[10,11]. Another advance in 3D bioprinting was the development of multimaterial printers, such as the integrated tissue-organ printer (ITOP), to impart the final constructs with mechanical strength and structural stability. This technology allows for the fabrication of structurally supported, cell-laden tissue constructs through sequential printing of cell-laden hydrogel of defined formulation along with a structural support polymer, such as polycaprolactone (PCL)[12]. The integration of PCL directly within the printing process enabled the ITOP to form stable and supported bioscaffolds on the human-scale.

Here, we develop a facile and efficient method to fabricate mechanically supported and anisotropic, multicomponent biomaterials, which employs surface tension forces to coat reticulated supports with cell-laden hydrogels. A 3D-printed framework is designed and produced with the desired geometry, structure, and mechanical properties. The reticulated scaffold is then dipped into a liquid hydrogel precursor solution (with or without cells) and wetting forces suspend liquid films across the windows as the material is withdrawn from the solution. The metastable, suspended liquid films are transformed into a stable network or hydrogel via standard crosslinking methods forming a permanent coating on the reticulated mesh. Importantly, the hydrogel coating on the outer surface of the construct enables facile encapsulation of mammalian cells as well as subsequent cell

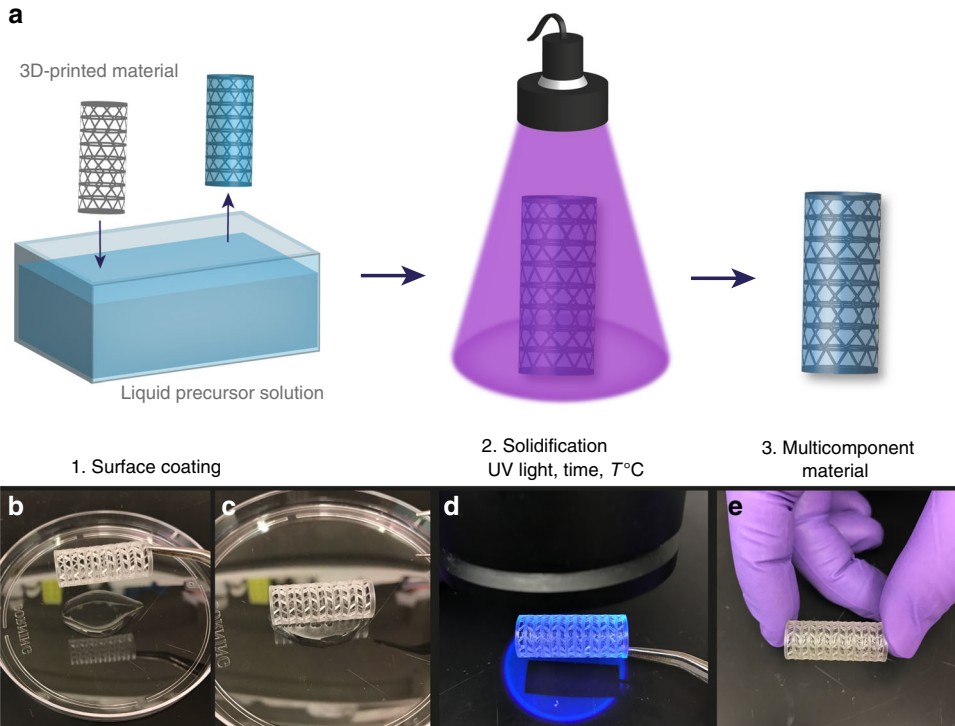

**Fig. 1** Surface tension-assisted additive manufacturing. **a** Schematic of the material fabrication process. **b** The 3D-printed reticulated scaffold is dipped into a liquid precursor solution. **c** Surface tension forces suspend a liquid film on the surface of the scaffold as the scaffold is withdrawn from the solution. **d** The suspended liquid film is crosslinked into a solid film by, for example, photopolymerization. **e** A stable solid coating is formed on the 3D-printed reticulated material

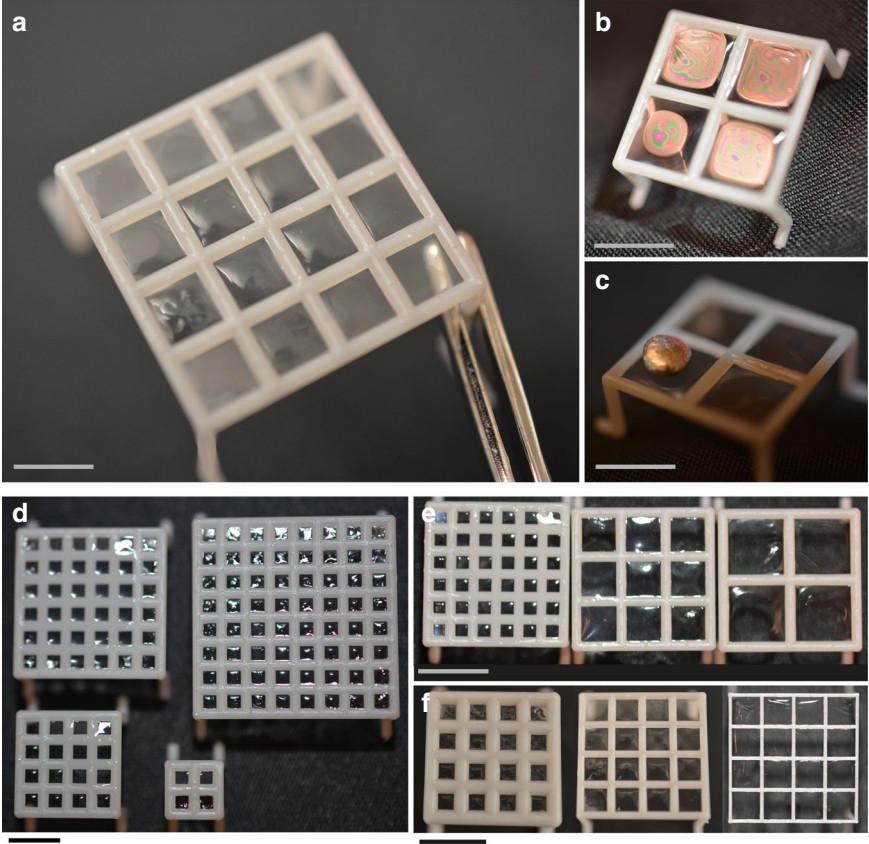

**Fig. 2** Tunable design parameters. **a** Flat scaffolds were coated with 7.5 wt% methacrylated gelatin that was photopolymerized (0.5 wt% LAP; $\lambda = 365$ nm; $I_0 = 6$ mW cm$^{-2}$; $t = 120$ s) to form solid films across the device windows. Scale bar, 2.5 mm. **b**, **c** After polymerization, a load-bearing film was formed across the device windows. Scale bars, 5 mm. **d** The coating process was scalable to larger surface areas (scaffolds of 0.5 × 0.5 cm, 1 cm × 1 cm, 1.5 cm × 1.5 cm, and 2 cm × 2 cm) in the same amount of fabrication time for the coating step. Scale bar, 5 mm. **e** Scaffolds with varied window sizes (2.25, 5.5, and 8.75 mm) were coated. Scale bar, 5 mm. **f**. Scaffolds with different pipe diameters (1, 0.5, and 0.25 mm) were coated. Scale bar, 5 mm

seeding. Mechanically and functionally complex biomaterials are made in this manner without the need to write material directly at every voxel in the final construct. This significantly decreases total fabrication time and minimizes cell handling. Furthermore, the process is versatile in material design and does not rely on any specialized equipment; the method is agnostic with respect to the upstream 3D printer, scaffolding material (e.g., resins, metals, biodegradable polymers), and class of hydrogel. Thus, this approach can potentially expand the regenerative medicine toolkit in a manner that is accessible to most biomedical labs.

## Results

**Surface tension-assisted additive manufacturing**. Surface tension-assisted additive manufacturing leverages surface wetting forces to suspend liquid films across the fenestrations of a reticulated mesh that can be transformed subsequently into a solid coating or hydrogel (Fig. 1). This approach allows the integration of a hydrogel with an engineered mechanical support, producing a final construct with both structural stability and a tailored cellular microenvironment.

3D printing was used to fabricate fenestrated mesh-like scaffolds with window spaces on the millimeter to centimeter length-scale (Fig. 1). The meshes were then dipped into a liquid precursor solution, e.g., polymer or protein solution, and wetting forces generated suspended liquid films across the fenestrations as the material was withdrawn from the solution. The suspended liquid films, present in the open windows of the 3D-printed

scaffolds, were then converted into solid films via an external trigger (temperature, time, light, or ionic gelation) forming a stable solid coating on the reticulated mesh, that covered the previously open windows. The method was validated initially with a methacrylated gelatin (7.5 wt%) coating on a planar mesh scaffold (Somos 9120; DSM). To solidify the suspended liquid films, we induced photopolymerization with low dose UV light (0.5 wt% lithium phenyl-2,4,6-trimethylbenzoylphosphinate (LAP); $\lambda = 365$ nm; $I_0 = 6$ mW cm$^{-2}$; $t = 120$ s) and uniform load-bearing gels were obtained in the windows of the scaffold (Fig. 2a–c). In this manner, the approach was not a surface coating in the traditional sense, wherein a thin film of a secondary material is deposited uniformly on an initial material, but a unique method to transform a reticulated mesh into an intact 3D and multicomponent object. The scale of device fabrication was easily increased to larger surface areas (0.5 cm × 0.5 cm to 2 cm × 2 cm devices shown here; Fig. 2d) without an increase in the coating time. In addition, the process was demonstrated for a range of window sizes and pipe diameters: square window lengths of 2.25, 5.5, and 8.75 mm (Fig. 2e) as well as scaffolds with pipes of 0.25, 0.5, and 1 mm in diameter (Fig. 2f).

**Model of suspended liquid film formation**. Liquids are known to suspend over open areas. Liquids will, for example, preferentially fill small voids in porous media rather than uniformly coat the struts even at relatively low volume fractions of liquid[13]. Recently, a similar phenomenon has been exploited to direct

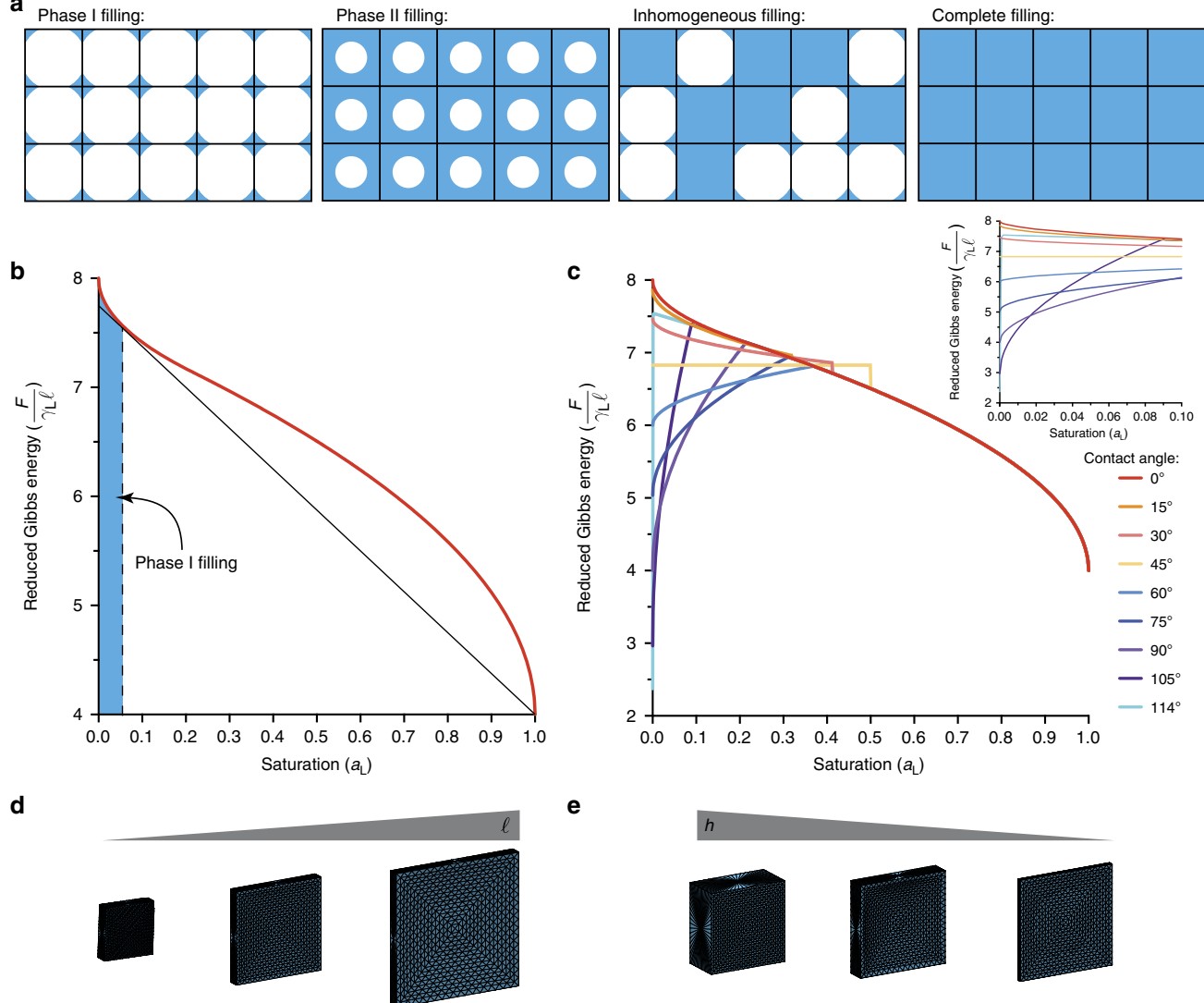

**Fig. 3** Suspended liquid film formation. **a** Phases of filling of a two-dimensional, square-cell mesh with increased liquid saturation. As described, phase II filling is not observed physically. **b** Plot of the reduced Gibbs energy as a function of saturation for $\theta = 0°$. **c** Plot of the reduced Gibbs energy as a function of saturation for $\theta = 0°$, 15°, 30°, 45°, 60°, 75°, 90°, 105°, and 114°. Inset graph for $\alpha_L = [0, 0.1]$. **d** Surface Evolver simulations of suspended liquid films with varying window lengths $l$. **e** Surface Evolver simulations of suspended liquid films with varying pipe diameters $h$

liquids for biomedical applications including the development of suspended microfluidics and spontaneous capillary flow[14]. An analogous physical phenomenon was exploited in our method to coat large area windows in engineered lattice frameworks. A simple model of suspended liquid film formation was developed to determine the physical constraints that allow for surface tension-assisted materials assembly. The model considered the Gibbs energy of liquid being added to a two-dimensional, square-cell mesh (Fig. 3a; Supplementary Figure 1). At small area fraction or saturation the liquid will preferentially wet the corners of each square cell to minimize its surface area, phase I filling (Supplementary Figure 2). The radius of curvature of the liquid meniscus or fillet is dictated by the contact angle, $\theta$, that the liquid makes with the scaffold material. The fillets should grow as the saturation increases and eventually touch, resulting in phase II filling (Supplementary Figure 3). Practical experience with, for example, wet wire screens indicates, however, that phase II filling is not observed. Instead, the liquid distribution undergoes a phase separation wherein some cells are completely filled and others remain only filled in the fillets, inhomogeneous filling. This

occurs for wetting liquids, $\theta < 90°$, as the overall Gibbs energy of the system is minimized and additional liquid serves to fill additional cells as opposed to increase the size of fillets in unfilled cells. Therefore, by submerging a reticulated scaffold in a wetting liquid, one can easily access the energetically favorable complete filling scenario, which is necessary for surface tension-assisted material assembly.

Our analysis employs a two-dimensional, square-cell mesh with a cell dimension, $l$, and the effect of gravity was ignored. A reduced Gibbs energy per cell as a function of contact angle, $\theta$, and saturation of the network, $\alpha_L$, for phase I and phase II filling were calculated from the geometry (refer to Modeling of suspended liquid films in the Supplementary Methods for a complete derivation).

$$\frac{F_I}{\gamma_L l} = 4\left(1 + \frac{\gamma_{SL}}{\gamma_L}\right) - (8 - 2\pi)\sqrt{\frac{a_L}{4 - \pi}}, \qquad (1)$$

$$\frac{F_{II}}{\gamma_L l} = 4\frac{\gamma_{SL}}{\gamma_L} + 2\sqrt{\pi}\sqrt{1 - a_L}. \qquad (2)$$

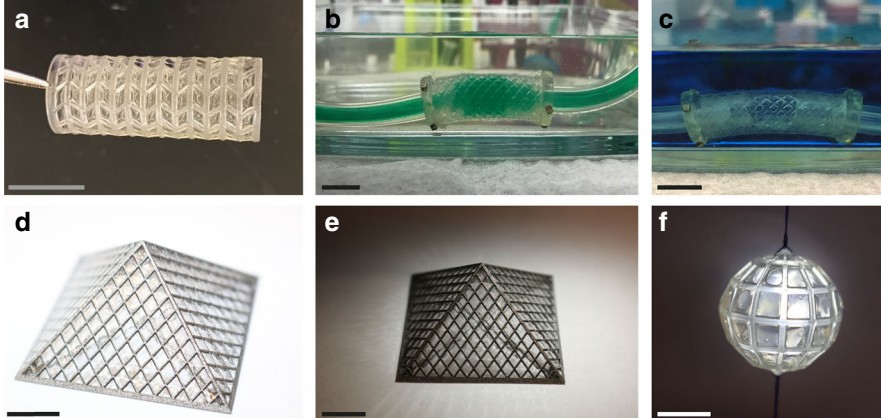

**Fig. 4** Material versatility and geometric control. **a** Tubular mesh structure coated with 2 mg mL$^{-1}$ collagen gel following 1 h gelation at 37 °C. Tubular meshes coated with 7.5 wt% methacrylated gelatin hydrogel formed a complete seal that held **b** liquid and **c** air pressure. **d**, **e** Louvre-like pyramids printed in stainless steel coated with neat thiol-ene networks via photopolymerization ($\lambda = 365$ nm; $I_0 = 6$ mW cm$^{-2}$; $t = 30$ s). **f**. Polyhedron coated with 7.5 wt% methacrylated gelatin. Scale bars, 1 cm

Here, $\gamma_L$ and $\gamma_{SL}$ are the surface energies of the liquid–vapor and solid–liquid interfaces, respectively. The surface energies are related to the contact angle, $\theta$, through Young's equation: $\gamma_S = \gamma_{SL} + \gamma_L \cos\theta$, where $\gamma_S$ is the surface energy of the solid–vapor interface. The saturation, $\alpha_L$, can be thought of as the fractional area of the open mesh filled with liquid. The angle subtended by the fillet constrains the critical saturation for phase I filling, which for $\theta = 0°$ corresponds to $a_{Lc} \approx 0.215$.

The observed coating phenomenon may be understood by inspecting the reduced Gibbs energy, $\frac{F}{\gamma_L p}$, as a function of saturation, $\alpha_L$, which was plotted for $\theta = 0°$ in Fig. 3b (a more complete analysis can be found in Supplementary Figures 4, 5, and 6). The observed change in sign of the second derivative of the reduced Gibbs energy with respect to saturation indicates a phase separation that corresponds to the transition from phase I filling to inhomogeneous filling. Further addition of liquid does not increase the volume of liquid in a given fillet but serves to increase the number of entirely filled cells. This is advantageous for surface tension-assisted materials assembly as complete filling is energetically favorable at high saturation, which is achieved by submersion in the liquid precursor solution.

A critical parameter in determining whether suspended liquid films will form on a reticulated mesh is the contact angle, $\theta$, that the liquid makes with the scaffold. This depends on both the liquid precursor solution and the solid material. Cases where complete coating with suspended liquid films is favorable also depends on the contact angle. The reduced Gibbs energy, $\frac{F}{\gamma_L p}$, as a function of saturation, $\alpha_L$, was plotted for a range of contact angles in Fig. 3c. Importantly, the reduced Gibbs energy for phase II filling is independent of $\theta$; however, $\alpha_{Lc}$ does depend on $\theta$. This analysis indicated that the liquid should be sufficiently wetting on the scaffold material to achieve a fully coated reticulated mesh, in other words, scaffold and precursor liquid pairs with low contact angle are easier to coat. Namely, the reduced Gibbs energy is lower for $a_L = 0$ than $a_L = 1$ for contact angles greater than 90° (Fig. 3c), implying that scaffold–liquid pairs with a large contact angle will prefer the uncoated state. This was also observed experimentally as hydrophobic meshes (Somos WaterShed XC11122, DSM) were difficult to coat completely using this method.

Additionally, the open source software package, Surface Evolver, was utilized to model suspended liquid films for square window geometries (Fig. 3d, e)[15]. Surface Evolver confirmed that

suspended liquid films form over a broad range of window lengths and pipe diameters (Fig. 3d, e).

**Material versatility and geometric control.** Surface tension-assisted additive manufacturing is not limited to a specific class of materials or additive manufacturing processes but can be used broadly in laboratories equipped with various 3D printers and working with different materials. Here, we utilized scaffolds made of various materials (i.e., metals, polymers), of different geometries, and printed with different additive manufacturing processes (i.e., powder bed fusion, vat photopolymerization) as demonstrated in Fig. 4 and Supplementary Figure 7. Therefore, by controlling the scaffold design and defining the coating properties, one can create a material with characteristics tailored for a given application.

The surface tension-assisted coating process enabled the fabrication of more complex scaffolds such as tubes and polyhedra as well as a Louvre-like pyramidal structure (Fig. 4). Metallic (stainless steel; direct metal laser sintering) and polymeric (Somos 9120, Accura®ClearVue; SLA) scaffolding materials were coated successfully. In addition, multiple liquid precursor solutions were employed to form solid films across scaffold fenestrations using different curing mechanisms (photopolymerization, temperature, and ionic gelation) and times ($t = 30$ s to 1 h). Protein (collagen and collagen-elastin), methacrylated gelatin, alginate, and neat thiol-ene networks were all used to coat the different geometries. The suspended liquid film was stable enough to allow for long polymerization times such as with collagen hydrogels (Fig. 4a). In addition, the coating process produced multicomponent materials with air and liquid tight seals (Fig. 4b, c).

**Structural support and mechanical anisotropy.** By using additive manufacturing and computer-aided design upstream, the final multicomponent material can be engineered to possess desired mechanical properties and anisotropy.

In physiology, many organs and tissues exhibit anisotropic mechanics, i.e., they possess different moduli along different axes, and this property is often difficult to recapitulate in traditional, monolithic biomaterials. The trachea for example demonstrates circumferential rigidity, imparted by sequential fibrocartilage rings, and longitudinal flexibility, provided by an elastic ECM

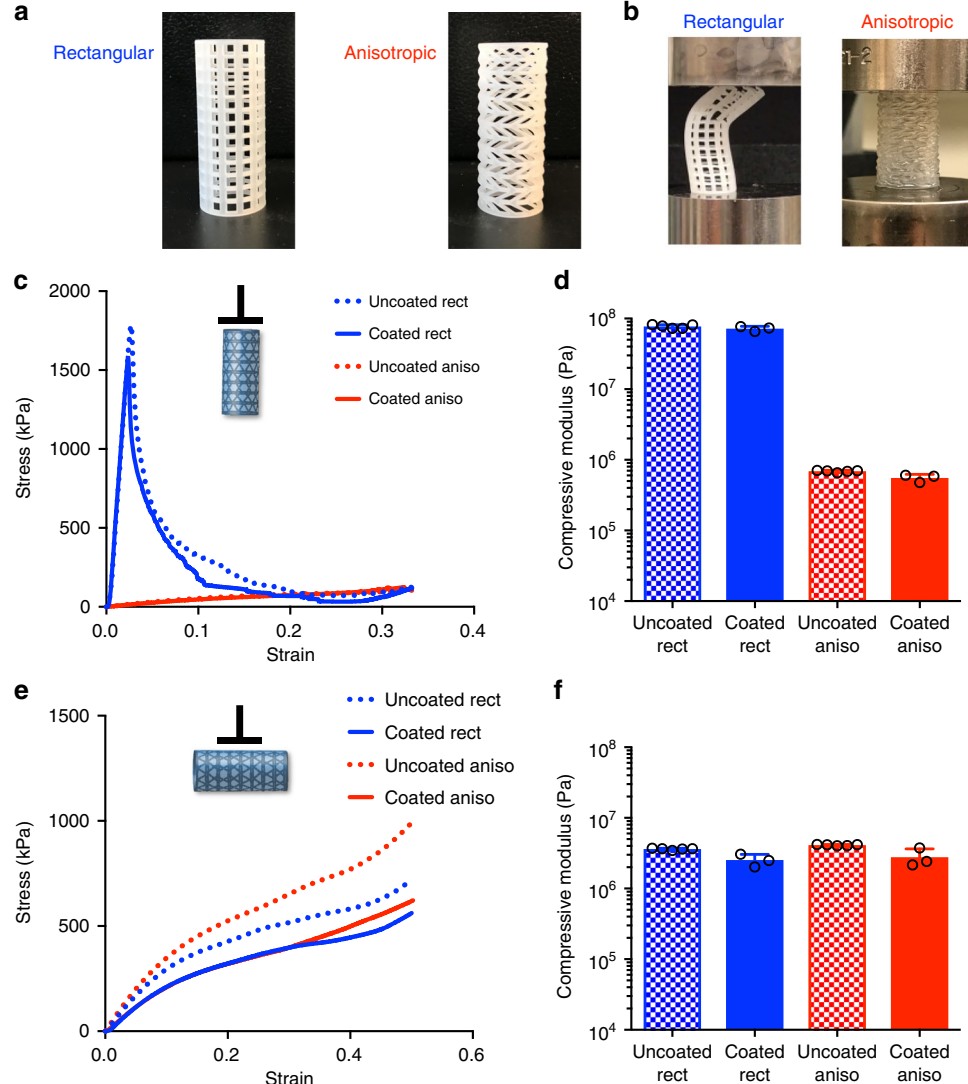

**Fig. 5** Mechanical characterization of cylindrical materials. **a** Two similar cylinders were produced via 3D printing with rectangular window geometry (rectangular; Rect) or parallelogram window geometry (anisotropic; Aniso). **b** Still images of the rectangular and anisotropic cylinders during longitudinal compression. Rectangular devices buckled and failed at low strain while anisotropic devices compressed up to ~30% strain without failure with or without a hydrogel coating (image shown with coating). **c** Stress–strain curves under longitudinal compression for uncoated (dashed lines) and coated (solid lines) rectangular (blue) and anisotropic (red) scaffolds. **d** Effective compressive moduli under longitudinal compression for uncoated (patterned bars) and coated (solid bars) rectangular (blue) and anisotropic (red) scaffolds. **e** Stress–strain curves under radial compression for uncoated (dashed lines) and coated (solid lines) rectangular (blue) and anisotropic (red) scaffolds. **f** Effective compressive moduli under radial compression for uncoated (patterned bars) and coated (solid bars) rectangular (blue) and anisotropic (red) scaffolds. The plots show a single representative stress–strain curve for each scaffold type. Compressive moduli values are displayed as mean + s.d. ($n = 5$ for uncoated scaffolds and n = 3 for coated scaffolds)

between the sequential rings[16]. This anisotropic structure is essential to tissue function as it allows the trachea to stretch or compress longitudinally during neck extension and adduction but prevents radial collapse during the pressure drops that occur while coughing or taking a deep breath[17,18].

The geometry of the windows of a cylindrical scaffold were designed as parallelograms such that the resulting anisotropic scaffold would compress easily along the longitudinal axis while maintaining radial rigidity (Fig. 5a). As a control, we designed a similar cylindrical scaffold with straight vertical lines, forming rectangular windows (Fig. 5a). The mechanical properties of each scaffold were compared under uniaxial compression along the longitudinal (Fig. 5b) and radial axes. Stress–strain curves confirmed that while both designs possessed similar compressive

moduli ($K = 3.6 \pm 0.4$ MPa and $4.1 \pm 0.3$ MPa for rectangular and anisotropic designs, respectively) in the radial test, they presented significantly different properties in the longitudinal dimension. The cylinders with rectangular windows exhibited a higher compressive modulus in the longitudinal test ($K = 77 \pm 2$ MPa) and buckled at relatively low strain ($\varepsilon = 2.6 \pm 0.1\%$), whereas the cylinders with the anisotropic design exhibited a bulk modulus two orders of magnitude lower ($K = 690 \pm 9$ kPa) and compressed reversibly to ~30% strain (Fig. 5c, d; Supplementary Figures 8, 9, and 10). Similar mechanical properties were observed when the scaffolds were coated with hydrogel (7.5 wt% methacrylated gelatin; Fig. 5c, e; solid lines). Importantly, the coating remained intact during the compression cycle without tearing or delaminating from the support. In these studies, design of experiments was not applied to

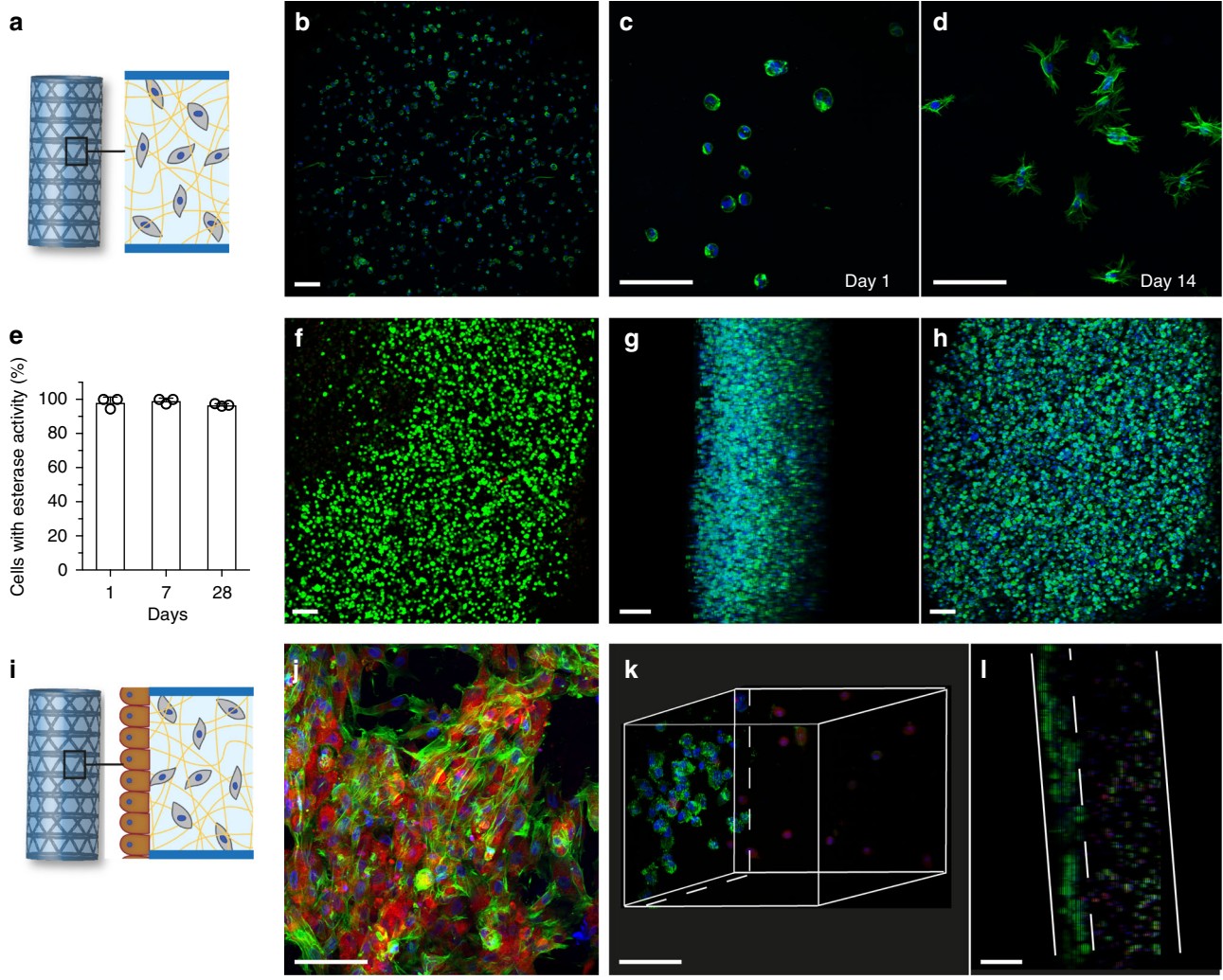

**Fig. 6** Cell-laden multicomponent scaffolds. **a** Schematic of cell encapsulation within the hydrogel coating of the multicomponent devices. **b** 475 µm Z-stack of a single, excised window 24 h after MRC-5 lung fibroblasts were encapsulated in 7.5 wt% methacrylated gelatin hydrogel coating ($1 \times 10^6$ cells mL$^{-1}$). Cell nucleus was labeled with NucBlue (blue) and actin cytoskeleton was labeled with Alexa Fluor$^{TM}$ 488 Phalloidin (green). **c** MRC-5 cells 1 day after encapsulation in 7.5 wt% methacrylated gelatin hydrogel coating ($1 \times 10^6$ cells mL$^{-1}$). Cell nucleus was labeled with NucBlue (blue) and actin cytoskeleton was labeled with Alexa Fluor$^{TM}$ 488 Phalloidin (green). Maximum intensity projection of 100 µm Z-stack. **d** MRC-5 cells spread in the gel 14 days after encapsulation. **e** Percentage of cells that displayed functional esterase activity as measured by live/dead assay at 1, 14, and 28 days following MRC-5 encapsulation in 7.5 wt% methacrylated gelatin coatings. Values are displayed as mean + s.d. ($n = 3$). **f** Live/dead staining of NHDF-laden hydrogel 1 day after encapsulation in 7.5 wt% methacrylated gelatin hydrogel coating at a high density ($10 \times 10^6$ cells mL$^{-1}$). 270 µm Z-stack of a single, excised window. **g** NHDF-laden hydrogel 1 day after encapsulation in 7.5 wt% methacrylated gelatin hydrogel at a high density ($10 \times 10^6$ cells mL$^{-1}$). Cell nucleus was labeled with NucBlue (blue) and actin cytoskeleton was labeled with Alexa Fluor$^{TM}$ 488 Phalloidin (green). Lateral view of a 388 µm Z-stack of a single, excised window. **h** Front view. **i** Schematic of co-culture scaffolds. **j** A monolayer of human mesenchymal stem cells (MSC) was seeded on the luminal surface of the tubular scaffold. Cell nucleus was labeled with NucBlue (blue), actin cytoskeleton was labeled with Alexa Fluor$^{TM}$ 488 Phalloidin (green), and cell cytoplasm was labeled with CellTracker Red (red). **k** NHDF were encapsulated within a gel and GFP-expressing HUVEC were seeded on the luminal surface of the scaffold. Cell nucleus was labeled with NucBlue (blue), actin cytoskeleton was labeled with Alexa Fluor$^{TM}$ 488 Phalloidin (green), and NHDF cell cytoplasm was labeled with CellTracker Red (red). 3D rendering of a 306 µm Z-stack of a single, excised window. **l** Lateral view of the co-culture scaffold. Dashed line indicates HUVEC layer on the luminal surface. Scale bars, 100 µm

isolate precise variation in mechanical properties caused by different processing conditions. Yet, it is clear that design and process influence the final mechanical properties of the device. A thorough investigation of the effects on each stage of the fabrication process on final part properties would be of interest but is beyond the scope of the current work.

**Biomaterial fabrication and tissue engineering**. The 3D-printed scaffold imparts the final constructs with defined mechanical

properties and the use of a hydrogel coating offers a support for cell seeding and/or encapsulation. As the coating process is amenable to a wide range of materials, hydrogel biophysical and biochemical composition can be tailored to the cell type and biological application of interest.

Cells can be encapsulated in the gel before coating (Fig. 6a) and/or cells can be seeded as a monolayer on the gel surface after crosslinking (Fig. 6i). Cylindrical scaffolds were coated with methacrylated gelatin encapsulating MRC-5 lung fibroblasts ($1 \times 10^6$ cells mL$^{-1}$ of gel). To ensure sterility, the scaffold was

autoclaved beforehand and the coating process and crosslinking were done in aseptic conditions. Confocal imaging of the scaffold windows demonstrated that MRC-5 fibroblasts were successfully encapsulated in the gel and spread over the course of two weeks (Fig. 6b–d). Cell viability, measured as esterase activity using live/dead assay remained above 95% 28 days after coating (Fig. 6e). In addition, the process was amenable to high cell density encapsulation and neonatal dermal fibroblasts (NHDF) were successfully encapsulated at a concentration of $10 \times 10^6$ cells mL$^{-1}$ of gel (Fig. 6f–h).

The method was then used for the rapid generation of co-culture systems with two different cell types. NHDF were encapsulated in the hydrogel, and a monolayer of human umbilical vein endothelial cells expressing GFP (HUVEC-GFP) was coated on the luminal surface of the tube (Fig. 6k, l). Similarly, a co-culture of encapsulated MRC-5 with a luminal layer of mesenchymal stem cells (MSC) was achieved (Fig. 6j).

## Discussion

Surface tension-assisted additive manufacturing of biomaterials enables simple and easy access to the fabrication of multicomponent materials by fusion of 3D-printed supports and hydrogel coatings. The coating is added to a 3D-printed scaffold in a subsequent step and the process respects the fundamental principle of additive manufacturing, which is to form 3D parts by the successive addition of material (ASTM/52900). The main innovation of this approach is to leverage surface tension to suspend stable liquid films across the fenestrations of a lattice after immersion in a hydrogel precursor solution. The liquid film is then crosslinked to generate solid windows that bear weight, hold pressure, and, in some cases, comprise a tunable niche for cell seeding and encapsulation. Although surface tension forces are well described, there is no report to our knowledge that exploits this physical phenomenon for the controlled fabrication of multicomponent (bio)materials in this manner. Importantly, the solid films not only modify the surface chemistry of the material but constitute an integral part of the engineered device itself. Traditional additive manufacturing, especially in the field of 3D bioprinting, has relied on material deposition or curing at each pixel independently within the final object, thereby coupling the overall time for part fabrication to the volume of the final part. In this work, we demonstrated that surface tension-driven coating can enable additive manufacturing of complex materials without the need for pixel-by-pixel fabrication and, here, the speed of the second stage of the process was volume invariant across the range of devices tested. Thus, the process not only increases the versatility of 3D bioprinting but also decreases the speed of fabrication through rational design. Further, this approach highlights that the final properties of a 3D-printed part do not rely solely on the material properties of the constituent materials but also on the design and fabrication of the part. Bio-integration, elasticity, lightness, and an intact seal can be imparted by the hydrogel coating while structural integrity and anisotropic mechanical properties are conferred by the 3D-printed backbone. Here, we focused on the coating of complex surfaces but this process could also be used, in principle, to fill the volume of other 3D-printed structures.

As the cell-laden hydrogel component is not formed during the 3D printing process, this approach avoids some of the challenges related to traditional 3D bioprinting, especially the difficulty of formulating bioinks that possess suitable rheological properties and biocompatibility as well as long cell handling times that can ultimately affect cell viability and function[19]. The surface tension-assisted coating is amenable to a large variety of liquid precursor solutions, and polymer coatings of tunable stiffness, composition, and/or chemical functionalization can be achieved depending on the cell type and the envisioned application. In this work, polymer,

protein, and polysaccharide-based hydrogels as well as non-swollen polymeric networks were employed as coatings. In the case of photopolymerization, the coating materials are clear solutions and there is negligible attenuation of the light throughout all structures tested and one should be able to fabricate structures that are micrometers to centimeters in thickness. The total dosage of UV light for the photopolymerization ($\lambda = 365$ nm; $t = 120$ s) was much lower than that for the post-processing of the 3D-printed devices ($\lambda = 405$ nm; 30 min). Further, we did not observe any significant change in the compressive moduli between the uncoated and coated devices. Therefore, modification of the mechanical properties of the parts during the photopolymerization of the coating was not a significant concern in this context but should be considered in future material fabrication.

In addition, the coating is gentle for cells and the rapid and scalable nature of the approach limits the total time that cells are exposed to adverse conditions. Lung fibroblasts that were encapsulated in the hydrogel showed an elongated and spread morphology over the course of 14 days and more than 95% viability was measured 28 days after encapsulation. The process is amenable to high cell density encapsulation as well as facile integration of different cell types in co-culture systems. In addition, there are different manners by which cells can be incorporated into the device: cells can be included in the gel precursor solution prior to dip coating and/or cells can be seeded as a monolayer on top of the gel after crosslinking, on the luminal or basal surface of the tube. Luminal cell monolayers of HUVEC or MSC were seeded on a fibroblast-laden hydrogel to generate physiologically relevant co-culture tissues. The different cell-loading possibilities expand the use of the method and the versatility of the type of cell-laden biomaterials and cell culture systems that one can create. Finally, the whole process can be completed under sterile conditions, as the 3D reticulated scaffold is autoclaved beforehand and the hydrogel coating done in aseptic conditions, hopefully easing future translation to in vivo implantation.

A major area of interest in tissue engineering is the fabrication of biomaterials with advanced mechanical properties that provide artificial tissues with biophysical properties that are tailored to the organ of interest and that are ultimately relevant to its biological function. As the reticulated scaffold is printed upstream, hierarchical structures can be engineered easily using the toolkit of CAD and additive manufacturing such that the final construct possesses the desired mechanical properties and anisotropy. For example, the mesh can be printed in a range of materials that provide rigidity or flexibility and designed with different window geometries (including squares, parallelograms, re-entrant honeycombs, rotating squares) that will impart tailored mechanics in one specific dimension (anisotropy) or specific physical properties (negative Poisson's ratio), which could be beneficial for biological function[20,21]. These physical characteristics are often difficult or impossible to recapitulate in a monolithic biomaterial, even with sophisticated 3D bioprinting technologies.

Here, we demonstrated that by tuning the window geometry we can fabricate scaffolds with anisotropic properties that compress longitudinally without damaging the hydrogel coating, better mimicking the mechanical needs of an engineered trachea. Other mechanical functionality could be envisioned and imparted to the scaffold by design, such as auxetic properties that would be beneficial for artificial blood vessels or skin substitutes. For example, re-entrant honeycombs and rotating squares as window microstructures could comprise auxetic, multicomponent materials[22,23]. In addition, the scaffold can be informed by advanced medical imaging to design implants with patient-specific geometry. Overall, this work highlights that the mechanical properties of an object are not controlled solely by constituent material properties but also by part design and processing.

Leveraging surface tension-assisted additive manufacturing, one can now fabricate materials with engineered properties tailored for a given application by designing the appropriate scaffold geometry and material as well as defining the coating. The facile and versatile nature of this method provides a flexible platform for various applications in tissue engineering and regenerative medicine, such as hollow biomaterials for organ replacement: trachea, nerve guides, and blood vessels grafts or substitutes could be engineered using this method, as well as flat structures for skin grafts and wound healing applications. Beyond the scope of tissue engineering, drug delivery systems or reservoirs, medical devices, as well as advanced in vitro cell culture systems could potentially be engineered by coating a hydrogel on a fenestrated scaffold to locally release drugs or mimic tissue structures to facilitate drug discovery and development, or study cellular interactions in a complex microenvironment.

## Methods

**Materials**. All reagents were purchased from Sigma-Aldrich and used as received, except as noted.

**Synthesis of methacrylated gelatin**. Methacrylated gelatin was synthesized as described previously[24]. Briefly, gelatin from porcine skin, type A, gel strength 300 bloom (20 g) was dissolved in deionized $H_2O$ (200 mL) at 50 °C under constant stirring for 60 min to obtain a 10 wt% gelatin solution. Methacrylic anhydride (12 g; 3.89 mmol $g^{-1}$ of gelatin) was added slowly to the gelatin solution, forming a homogeneous suspension. The suspension was reacted at 50 °C for 1.5 h. After reaction, the suspension was transferred to 50 mL conical tubes, centrifuged (3500 rcf for 5 min), and the supernatant was decanted to recover the methacrylated gelatin solution. The methacrylated gelatin solution was diluted with two volumes of deionized $H_2O$ (40 °C) and transferred to dialysis tubing (Spectra/Por 7 MWCO 10000; Spectrum Laboratories). The methacrylated gelatin solution was dialyzed for 5 days at 40 °C to remove unreacted methacrylic anhydride. After dialysis, the pH of the solution was brought to ~7.4 with 1 N NaHCO₃ and lyophilized to recover methacrylated gelatin as a foam. The methacrylated gelatin was resuspended in UltraPure™ Distilled Water (37 °C; Invitrogen) to 10 wt% and stored at 4 °C until use. The degree of functionalization was determined to be ~80 % by a ninhydrin assay comparing to unmodified gelatin.

**Lithium phenyl-2,4,6-trimethylbenzoylphosphinate synthesis**. Lithium phenyl-2,4,6-trimethylbenzoylphosphinate (LAP) was synthesized as described previously[25,26]. At room temperature and under argon, 3.2 g (0.018 mol) of 2,4,6-trimethylbenzoyl chloride was added slowly to an equimolar amount of dimethyl phenylphosphonite (3 g) under stirring. The mixture was reacted for 18 h at room temperature and then lithium bromide (6.1 g; 4 eq.) in 100 mL of 2-butanone was added. The reaction was then heated to 50 °C and, after 10 min, a solid precipitate formed. The mixture was cooled to room temperature, allowed to rest for 4 h, and then filtered. The filtrate was washed and filtered 3 times with 2-butanone to remove unreacted lithium bromide and excess solvent was removed under vacuum. The product, lithium phenyl-2,4,6-trimethylbenzoylphosphinate (LAP) was recovered in near quantitative yields ¹H NMR (Bruker Avance QNP 400; 400 MHz, D₂O, d): 7.57 (m, 2H), 7.42 (m, 1H), 7.33 (m, 2H), 6.74 (s, 2H), 2.09 (s, 3H), 1.88 (s, 6H).

**Additive manufacturing of reticulated scaffolds**. All scaffolds were designed in AutoCAD® software (Autodesk, San Rafael, CA, USA). The metal pyramids were fabricated by powder bed fusion (laser power 90 W) using a ConceptLaser ML2 DMLS printer (Proto Labs, Maple Plain, MN, USA) in stainless steel 17–4 PH (fineline™, Rayleigh, NC, USA). The print orientation was such that the base of the pyramid was built on the build plate with the apex directly above the centroid of the base. As post-processing, the metal parts were cleaned of excess powder and subjected to a stress relief heat cycle and H900 heat treatment. The parts were then cut from supports manually, sanded off, grit blasted and shot with glass beads. The polymeric scaffolds were fabricated by vat photopolymerization using Viper SLA or Projet 6000 3D printers (3D Systems, Rock Hill, SC, USA). The mesh squares and polyhedron were printed in Somos 9120 (DSM, Heerlen, The Netherlands). The tubular scaffolds used for cell culture experiments were printed in the biocompatible Accura®ClearVue (3D Systems). Laser diameter was 100 μm and parts were printed in 0.004″ (~100 μm) layers. As post-processing, the polymeric parts were cleaned of excess resin with isopropyl alcohol, dried, and UV cured for 30 min. Supports were manually taken off, the parts were sanded to remove support marks, and grit blasted. To ensure sterility, the scaffolds were autoclaved prior to cell seeding. A full description of the 3D-printed geometries is provided in Supplementary Table 1.

**Surface tension-assisted materials assembly**. The procedure for fabricating multicomponent materials via surface tension-assisted additive manufacturing employs the upstream 3D-printed mesh scaffold as well as the preparation of the coating precursor solution. The process is similar regardless of the nature of the coating and type of scaffold: the 3D-printed scaffold is dipped into a pool of the gel precursor solution, ensuring that all windows are coated uniformly, and cross-linking is induced to transform the suspended liquid film into a solid film. As opposed to a traditional surface coating whereby a thin film of a secondary material is deposited uniformly on a surface, this method transforms the mesh into a 3D and multicomponent object by generating solid windows within the 3D-printed scaffold. Versatility of the coating material and polymerization method was demonstrated using four different coating materials (methacrylated gelatin hydrogels, neat thiol-ene networks, collagen hydrogels, and alginate hydrogels). However, for the rest of the study (i.e., influence of scaffold design parameters, mechanical testing study, and cell culture experiments) methacrylated gelatin coatings were employed.

**Formation of methacrylated gelatin coatings**. The hydrogel precursor solution was prepared by mixing pre-heated (40 °C) methacrylated gelatin solution (10 wt% in UltraPure® distilled water) with LAP (5 wt% in UltraPure® distilled water) to final concentrations of 5 to 7.5 wt% methacrylated gelatin and 0.5 wt% LAP. The balance of the solutions was composed of UltraPure® distilled water. Photo-polymerization was employed to crosslink the methacrylated gelatin solution with an LED-based UV lamp ($\lambda = 365$ nm; $I_0 = 6$ mW cm$^{-2}$; $t = 120$ s; Thorlabs). During irradiation, LAP generates radicals that initiate polymerization of the methacrylate groups of the methacrylated gelatin, ultimately, percolating a network within the suspended liquid film and inducing hydrogel formation constituting a solid window[24,26].

**Formation of thiol-ene coatings**. Metal pyramid scaffolds were coated with two different neat thiol-ene coatings. In one set of devices, trimethylolpropane tris(3-mercaptoproprionate) (3-SH) was reacted with allyl ether (2-ENE) on stoichiometry. Here, a precursor solution of 3-SH (2826 mg; 21.3 mmol-SH), 2-ENE (1045 mg; 21.3 mmol-ENE), and benzophenone (25 mg; 0.14 mmol) was prepared. In another set of devices, pentaerythritol tetrakis(3-mercaptoproprionate) (4-SH) was reacted with pentaerythritol allyl ether (4-ENE) and allyl glycidyl ether (ENE). Here, a precursor solution of 4-SH (3 g), 4-ENE (828 μL), ENE (138 μL), and 2,2-dimethoxy-2-phenylacetophenone (39 mg) was prepared. In each case, after coating the scaffold with the thiol-ene precursor, crosslinking was induced with an LED UV lamp ($\lambda = 365$ nm; $I_0 = 6$ mW cm$^{-2}$; $t = 30$ s; Thorlabs).

**Formation of collagen and alginate coatings**. Collagen (collagen 1 from rat tail, Corning) hydrogel precursor was prepared following manufacturer's instructions at a final collagen concentration of 2 mg mL$^{-1}$. After coating, gelation was achieved by placing the coating device in a sterile incubator (37 °C) for 1 h. Alginate hydrogel precursor was prepared by mixing sodium alginate (Novamatrix) in UltraPure® distilled water at 1.4 wt%. After coating, the suspended liquid films were crosslinked following a secondary immersion in a 20 mM BaCl₂ solution.

**Numerical simulation of suspended liquid films**. Surface Evolver software was employed to perform numerical simulations on the stability of suspended liquid films[15]. Surface Evolver is an established, open source software package for studying equilibrium fluidic surfaces that uses finite element methods with a gradient descent method to calculate the minimal energy state of fluid–fluid and fluid–solid interfaces. For this work, single square window geometries were tested for suspended liquid film stability. The square window was filled with a volume of liquid and Surface Evolver was run to energetic-equilibrium to determine film stability. If energetic-equilibrium could not be reached, it was determined that the suspended liquid film was not stable for the given conditions.

**Mechanical testing**. Coated and uncoated scaffolds with both a rectangular and anisotropic window design were mechanically tested under uniaxial compression along the longitudinal and radial axes. For all tests, an Instron 5943 testing machine was employed with a 500-N load cell and steel compression platens. Specimens were subjected to displacement at a rate of 3 mm min$^{-1}$ with a total displacement of 10 mm and 6 mm in the longitudinal and radial tests, respectively. Force was converted into stress ($\sigma = F/SA$, where $F$ was the force and SA was the cross-sectional surface area in the test geometry) and displacement was converted into strain ($\varepsilon = \Delta L/L_0$, where $\Delta L$ was the displacement and $L_0$ was the initial length in the test geometry). An effective modulus under uniaxial compression ($K$) was determined for each sample as the slope of the linear fit of the stress–strain curve over the first 0.5 mm of compression. The experiments were planned with two factors: geometry and coating. Each factor had two levels—geometry: rectangular or anisotropic; coating: uncoated or coated. Measurements of the rectangular devices (uncoated and coated) were performed on $n = 5$ samples and measurements of the anisotropic devices (uncoated and coated) were performed on $n = 3$ samples.

**Cell-laden materials and confocal imaging**. MRC-5 lung fibroblasts (ATCC) were cultured in alpha-MEM supplemented with 10% fetal bovine serum (FBS) and 1% penicillin–streptomycin. Neonatal dermal fibroblasts (NHDF, ATCC) were cultured in alpha-MEM supplemented with 15% FBS and 1% penicillin–streptomycin. Human mesenchymal stem cells (MSC, Texas A&M University System Health Science Center) were cultured in alpha-MEM supplemented with 20% FBS and 1% penicillin–streptomycin. Human umbilical vein endothelial cells that express the Green Fluorescence Protein (HUVEC-GFP, Kamm Lab/MIT) were cultured on a collagen 1-coated plate (50 µg mL$^{-1}$, Corning) in EGM2 medium (Lonza) supplemented with 10% FBS and 1% penicillin–streptomycin. Cells were passaged when they reached 70–80% confluence. All cell lines and primary cells tested negative for mycoplasma contamination.

MRC-5 cells were encapsulated in methacrylated gelatin hydrogel at a concentration of $1 \times 10^6$ cells mL$^{-1}$ of gel. Live/dead assay (Life technologies) was performed 1, 7, and 28 days after encapsulation, following manufacturer's instructions. Viability percentages were obtained by counting the number of viable, green fluorescent cells, and dividing by the total cell number. The counting was performed on three independent hydrogel windows. For the high cell density experiment, NHDF were encapsulated in methacrylated gelatin hydrogel at a concentration of $10 \times 10^6$ cells mL$^{-1}$ of gel. Live/dead assay and actin staining were performed 24 h after cell encapsulation.

For the co-culture experiment, NHDF were stained with CellTracker Red (Life Technologies) for 35 min at 1 µM and then encapsulated in methacrylated gelatin hydrogel at a concentration of $1 \times 10^6$ cells mL$^{-1}$. HUVEC-GFP ($1.5 \times 10^6$ mL$^{-1}$) were seeded in the scaffold luminal surface the following day. To improve HUVEC attachment, the scaffold lumen was coated with collagen (type 1, rat tail; 50 µg mL$^{-1}$ Corning) for 20 min prior cell seeding.

Before confocal imaging, the scaffolds with cells were fixed with 3.2% paraformaldehyde (PFA) for 15 min at room temperature and cell nuclei and actin filaments were stained with NucBlue (Thermofisher Scientific) and Alexa Fluor® 488 Phalloidin (Life Technologies, Carlsbad, CA) respectively, for 2 h at 4 °C. Olympus FV1200 confocal microscope with a ×10 objective and a ×30 oil immersion objective were used for imaging and image reconstruction was done using Image J.

The experiments were performed on $n = 10$ scaffolds for MRC-5 encapsulation, $n = 5$ for co-culture, and $n = 3$ for high cell density encapsulation.

**Data availability**. All relevant data included in this manuscript are available from the authors.

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

## Acknowledgements

We thank R.D. Kamm for gifting the HUVEC-GFP cells and E.R. Dufresne for constructive feedback during the preparation of this manuscript. This work was supported by the Leona M. and Harry B. Helmsley charitable trust. M.W.T. thanks the NIH for funding support through a Ruth L. Kirschstein National Research Service Award (F32HL122009).

## Author contributions

H.R., M.W.T., and R.L. conceived the project; H.R., M.W.T., D.G.A., M.J.C., and R.L. designed the experiments; H.R., M.W.T., S.-Y.W., and M.A.C. designed the 3D-printed scaffolds; G.Z.C. and S.P.G. contributed to the design of tubular scaffolds; H.R., M.W.T., and S.-Y.W. conducted all experiments; H.R., M.W.T., and M.J.C. developed the physical model; H.R., M.W.T., S.-Y.W., D.G.A., M.J.C., and R.L. analyzed and interpreted the results; H.R. and M.W.T. wrote the manuscript, with significant contributions from all authors. All authors approved the final version of the manuscript.

## Additional information

**Competing interests:** The authors declare no competing interests.

