## [Peer Review File(PDF 433 kb) · Nature Communications]

Reviewers' Comments:

Reviewer #1:

Remarks to the Author:

The paper contain some new and significant scientific information adequate to justify publication.

However the reporting of the study need to be improved. The reporting of the experimental methods and results should be more complete and accurate.

Abstract does not contain any quantitate data or clear results, only method.

Literature part there should be more relevant references related to curing AM parts and also what has been already done in UV curing industry to clarify the novelty. Example references to read: Kao, C. T., Lin, C. C., Chen, Y. W., Yeh, C. H., Fang, H. Y., & Shie, M. Y. (2015). Poly (dopamine) coating of 3D printed poly (lactic acid) scaffolds for bone tissue engineering. *Materials Science and Engineering: C*, 56, 165-173.

Landers, R., Hübner, U., Schmelzeisen, R., & Mülhaupt, R. (2002). Rapid prototyping of scaffolds derived from thermoreversible hydrogels and tailored for applications in tissue engineering. *Biomaterials*, 23(23), 4437-4447.

Scott, M. L. (1985). A review of UV coating material properties. In *Laser Induced Damage In Optical Materials*: 1983. ASTM International.

Major claim is a coating process for AM parts (scaffolds). However, it should be more clearly stated what is new in the research.

The test parts made with parameters should be gathered in table format. Now text is quite hard to read.

In AM I recommend to use ISO / ASTM52900 – 15 Standard Terminology for Additive Manufacturing – General Principles – Terminology. For the sake of clarity and for future understandability and indexing when standard name overrules other.

Since process parameters have remarkable effect on properties of AM parts, what was the process parameters used in different processes: in Vat photopolymerization and Powder bed fusion with the different materials. How about the post processing of the parts? How was the supports removed? What kind of finishing was used? Thermal and/or UV curing? Heat treatments for metals?

Do the UV also affect to the AM parts? Since some materials were UV curable, more UV will ageing them. How material properties was changed?

What was a reasoning to select certain AM processes? Why 3 different materials? How not only focusing on one?

About the design of the parts: What was the all design parameters used? Now there seems to be only size of the "window". Thickness of the of parts and pipes? What about the 3D form shown in Figure 1 & 4?

About Mechanical characterization. Amount of samples is quite small. How much variation there was between measurements? What was the repeatability?

Explain in detail how the experiments of the samples have been planned and analyzed (factors, levels, type of experimental plan, replications, analysis of variance and related statistical tests. Was there predefined experimental design, e.g. a factorial plan? Explain how the process

variables(if tested) influence the responses (individual effects and possible interactions).

About the cell cultures, how many samples there were? Parallel and reference samples?

When thinking geometry of the part, how very complex parts with inner structures example pipes could be coated? Will the curing penetrate? With dipping hot to get and make sure that existing resin is get out?

I had a feeling that too many things (different AM process, materials etc.) has been reported in same paper. I would encourage to deeply think what are the most important findings and limit reporting experiment not related to that. At the moment paper is quite hard to read and takes lot of time to find what sample is related to what test or experiment.

I recommend Major revision for the paper to see more about the details and novelty of the submission.

Reviewer #2:

Remarks to the Author:

The paper describes a new method for coating 3D printed reticulated scaffolds with a polymerisable liquid film, through surface tension-driven filling of the fenestrations in the scaffolds.

The phenomenon is explained through modelling the Gibbs free energy with increasing saturation, for a number of different contact angles and ratios of solid-liquid to solid-vapour surface energies. Examples of coated structures are shown, including a tracheal scaffold that could be optimised for anisotropic mechanical behaviour independent of the coating. The examples also include scaffolds coated with hydrogels encapsulating cells of different types, in cases followed by seeding of a second cell type on the coating surface.

Although the described phenomena must have occurred before in tissue engineering research, to my knowledge this is the first report of it being exploited for the controlled coating and cell seeding of scaffolds.

It is unclear to me though if it works only for the outer surface, or throughout the bulk of the scaffold (fenestrations within). This is not stated clearly, and some remarks seem contradictory on this point, e.g. 'enables facile encapsulation of mammalian cells throughout the full three-dimensional construct' vs. 'various applications (...) such as hollow biomaterials (...) as well as flat structures'. This should be clarified.

The detailed explanation of the phenomenon using a physical chemistry model strengthens the work. However, the manuscript lacks explanation on how the graphs in Fig. 3 should be interpreted; e.g. is there a max. contact angle for which the technique will work, and if so how can that be seen from the plot?

A major shortcoming is that despite tissue engineering being named as the main application, none of the used scaffold materials is biodegradable. Why is this only demonstrated for a metal and a number of densely crosslinked, non-resorbable stereolithography resins? It may only be for convenience, but it begs the question if it would work just as well for more relevant materials (i.e. biomaterials), which may be 3D printed into scaffolds with lower accuracy, different surface roughness, different contact angles etc.

Ideally it would be demonstrated for more relevant scaffolds; if not it should at least be justified why it would work just as well.

When dip coating a scaffold into a cell suspension, I can imagine there would be a considerable dead volume of suspension that is lost, resulting in a low seeding efficiency. Please elaborate.

Finally, I don't think the technique should be termed 'Surface tension-assisted additive manufacturing', as the coating is performed after the preparation of a scaffold by additive manufacturing, and the coating procedure does not have the characteristics of additive manufacturing (as are set out in ASTM standards).

Reviewer #3:

Remarks to the Author:

This research is well reported and elegant. The dual step technique to address the problem of 3D printing with cells for tissue engineering applications is key. Using the advantages of 3D printing for producing lattice structures and the physical nature of surface tension in these structures to fill in the holes with a secondary material is clever. Applying mechanical design to the lattice hole shapes and sizes to control the mechanical properties and anisotropy of the produced object is a great opportunity to communicate with the wider audiences of Nature. It may allow readers to truly understand that it is not just material properties that control mechanical properties but also the design and process.

Reviewer #1 (Remarks to the Author):

The paper contain some new and significant scientific information adequate to justify publication.

However the reporting of the study need to be improved. The reporting of the experimental methods and results should be more complete and accurate.

We are happy to read that the reviewer felt that the paper contained “significant scientific information adequate to justify publication.” We thank the reviewer for his or her helpful comments, which are addressed specifically below, and we feel that by revising the manuscript accordingly we have produced a more clear and thorough presentation of our findings.

1. Abstract does not contain any quantitate data or clear results, only method.

We agree that the abstract would benefit from a revision. We have modified it accordingly following the reviewer’s suggestion to include quantitative data and clear results.

“The proliferation of computer-aided design and additive manufacturing enables on-demand fabrication of complex, three-dimensional structures. However, combining the versatility of cell-laden hydrogels within the 3D printing process remains a challenge. Herein, we describe a facile and versatile method that integrates polymer networks (including hydrogels) with 3D printed mechanical supports to fabricate multicomponent (bio)materials. The approach exploits surface tension to coat fenestrated surfaces with suspended liquid films that can be transformed into solid films. The operating parameters for the process are determined using a physical model, and complex geometric structures are successfully fabricated. We engineer, by tailoring the window geometry, scaffolds with anisotropic mechanical properties that compress longitudinally (~30% strain) without damaging the hydrogel coating. Finally, the process is amenable to high cell density encapsulation and co-culture. Viability (>95%) was maintained 28 days after encapsulation. This general approach can generate biocompatible, macroscale devices with structural integrity and anisotropic mechanical properties.”

2. Literature part there should be more relevant references related to curing AM parts and also what has been already done in UV curing industry to clarify the novelty.

Example references to read:

Kao, C. T., Lin, C. C., Chen, Y. W., Yeh, C. H., Fang, H. Y., & Shie, M. Y. (2015). Poly (dopamine) coating of 3D printed poly (lactic acid) scaffolds for bone tissue engineering. *Materials Science and Engineering: C*, 56, 165-173.

Landers, R., Hübner, U., Schmelzeisen, R., & Mülhaupt, R. (2002). Rapid prototyping of scaffolds derived from thermoreversible hydrogels and tailored for applications in tissue engineering. *Biomaterials*, 23(23), 4437-4447.

Scott, M. L. (1985). A review of UV coating material properties. In *Laser Induced Damage*

In Optical Materials: 1983. ASTM International.

We thank the reviewer for providing these references and we have included reference to the Kao and Landers papers in the revised manuscript.

Introduction section, page 2:

“Embedded bioprinting approaches have been developed wherein material is printed directly into a temporary support bath. Examples include a shear-thinning hydrogel or a thermosensitive gelatin microparticle suspension that provides a temporary mechanical support during printing and is removed after crosslinking, enabling omni-directional printing.[8,9]. This technique has enabled the fabrication of elegant and complex soft materials. In other work, polylactic acid or hydrogel-based scaffolds were printed with micron-scale pores and immersed in a coating solution post-fabrication. The coatings improved cell adhesion and biointegration of the 3D printed structures.[10,11]”

- [10] R. Landers, U. Hubner, R. Schmelzeisen, R. Mulhaupt, Rapid prototyping of scaffolds derived from thermoreversible hydrogels and tailored for applications in tissue engineering., *Biomaterials*. 23 (2002) 4437–4447.
- [11] C.-T. Kao, C.-C. Lin, Y.-W. Chen, C.-H. Yeh, H.-Y. Fang, M.-Y. Shie, Poly(dopamine) coating of 3D printed poly(lactic acid) scaffolds for bone tissue engineering., *Mater. Sci. Eng. C. Mater. Biol. Appl.* 56 (2015) 165–173.

The Scott reference is not pertinent to the current manuscript as we are using a fundamentally different approach than the UV coatings mentioned in the Scott reference. Here, we employ low-dose UV light to solidify a suspended liquid film in the open windows of a 3D printed scaffold. This is not purely a surface coating in the traditional sense as discussed by Scott, wherein a thin film of a secondary material is deposited uniformly on an initial material, but a unique method to transform a reticulated mesh into a complete three-dimensional and multicomponent object.

3. Major claim is a coating process for AM parts (scaffolds). However, it should be more clearly stated what is new in the research.

The novelty of the research is to leverage surface tension forces to suspend liquid films in the windows of fenestrated structures. Uniquely, the suspended liquid films are transformed into a solid material, often with UV light, and the surface tension coating and solidification is part of constructing the final design. The coated films are not only to modify the surface chemistry of the material but part of the engineered device itself. Although surface tension forces are well described, there is no report to our knowledge of using this physical phenomenon for the controlled fabrication of multicomponent (bio)materials. In the above references mentioned by the reviewer, filling of porous scaffolds was achieved, but the scaffold pore size was small (400 or 500 μm), no evidence for larger window size was demonstrated, and there is no mention of surface tension wetting. In addition, in one of the examples, there is no subsequent polymerization, just an immersion and adsorption of the liquid on the mesh as is common in traditional coating applications. Further, traditional additive manufacturing, especially that in the field of 3D bioprinting, has relied on material deposition or curing at each pixel independently within the final object, thereby coupling the overall time for part fabrication to the volume of the final part. In this work, we uniquely demonstrate that exploiting surface tension-driven coating can enable additive manufacturing of complex, multicomponent materials without the need for pixel-by-pixel fabrication and, here, the speed of the second stage of the process is volume

invariant. Thus, the process not only increases the versatility of 3D printing and 3D bioprinting but also presents methods to decrease the speed of fabrication through rational design. Finally, this work demonstrates that the final properties of a 3D-printed part do not rely solely on the constituent material properties but also on the design and fabrication of the part.

To clarify the novelty of the research, we have added the following sentences in the Discussion section (page 13):

“The novelty of this approach is to leverage surface tension forces to suspend stable liquid films across the fenestrations of a lattice after dip coating into a hydrogel precursor solution. The liquid film is then crosslinked, generating solid films that bear weight, hold pressure, and constitute a tunable niche for cell seeding and encapsulation. Although surface tension forces are well described, there is no report to our knowledge that exploits this physical phenomenon for the controlled fabrication of multicomponent (bio)materials. Further, in this work, the solid films not only modify the surface chemistry of the material but form part of the engineered device itself. Bio-integration, elasticity, lightness, and an intact seal can be imparted by the hydrogel coating while structural integrity and anisotropic mechanical properties are conferred by the 3D printed backbone.”

4. The test parts made with parameters should be gathered in table format. Now text is quite hard to read.

We thank the reviewer for this helpful suggestion and we agree that a table presenting part characteristics would improve the clarity of the manuscript. We have added the following Table 1 to the Supplementary Information, which summarizes the design parameters as well as the materials of the different 3D parts used in the study.

Supplementary Table 1: Design parameters of the 3D-printed scaffolds

Scaffold geometry	Scaffold material	Dimensions (cm)	Window size [#] (mm)	Pipe diameter (mm)
Mesh squares	Somos 9120	0.5 x 0.5	2.25 x 2.25	0.5
		1 x 1		
		1.5 x 1.5		
		2 x 2	2.25 x 2.25	0.5
		1 x 1	5.5 x 5.5	
			8.75 x 8.75	
		N.D.	0.25	
		1 x 1	0.5	
			1	
Polyhedron	Somos 9120	2 x 2 (d x h)	N.D.	0.5
Pyramid	Stainless steel 17-4PH	4 x 2 (a x h)	N.D.	0.5
Tubular scaffold	AccuraClearVue	3 x 1 (l x d)	N.D.	0.5

a: base length; d: diameter; h: height; l: length; N.D.: not determined; #: window size was calculated as length x width for center-to-center distance between adjacent struts

5. In AM I recommend to use ISO / ASTM52900 – 15 Standard Terminology for Additive Manufacturing – General Principles – Terminology. For the sake of clarity and for future understandability and indexing when standard name overrules other.

We thank the reviewer for this advice and we agree that using ISO/ASTM52900 terminology will help improve clarity and indexing of the manuscript. We have now included the the following terms as defined by ISO standards in the manuscript:

“vat photopolymerization” (page 16); “powder bed fusion” (page 16); “curing” (page 9); “fusion” (page 13).

6. Since process parameters have remarkable effect on properties of AM parts, what was the process parameters used in different processes: in Vat photopolymerization and Powder bed fusion with the different materials. How about the post processing of the parts? How was the supports removed? What kind of finishing was used? Thermal and/or UV curing? Heat treatments for metals?

Based on the reviewer's suggestion, we have added the following clarifications in the Method section regarding the AM processes (pages 16-17):

“All scaffolds were designed in AutoCAD® software (Autodesk, San Rafael, CA, USA). The metal pyramid was fabricated by powder bed fusion using a ConceptLaser ML2 DMLS printer (Proto Labs, Maple Plain, MN, USA) in Stainless steel 17-4 PH (fineline™, Raleigh, NC, USA). As post-processing, the metal parts were cleaned of excess powder and subjected to a stress relief heat cycle and H900 heat treatment. The parts were then cut from supports manually, sanded off, grit blasted, and shot with glass beads. The polymeric scaffolds were fabricated by vat photopolymerization using Viper SLA or Projet 6000 3D printers (3D Systems, Rock Hill, SC, USA). The mesh squares and polyhedron were printed in Somos 9120 (DSM, Heerlen, The Netherlands). The tubular scaffolds used for cell culture experiments were printed in the biocompatible Accura®ClearVue (3D Systems). Laser diameter was 100 μm and parts were printed in 0.004” (~100 μm) layers. As post-processing, the polymeric parts were cleaned of excess resin with isopropyl alcohol, dried, and UV cured for 30 min. Supports were manually taken off, the parts were sanded to remove support marks, and grit blasted.”

7. Do the UV also affect to the AM parts? Since some materials were UV curable, more UV will ageing them. How material properties was changed?

We did not test rigorously how the material properties of the AM parts were changed by photopolymerization of the hydrogel. The dosage used for the photopolymerization of the hydrogel or thiol-ene networks ($I_0 = \sim 6.0 \text{ mW}\cdot\text{cm}^{-2}$; $t = 120 \text{ sec}$) was much lower than that used for the UV post-processing of the 3D-printed devices, which was 30 min. In addition, the photopolymerization of the hydrogel films used a lower wavelength light ($\lambda = 365 \text{ nm}$) compared to UV curing ($\lambda = 405 \text{ nm}$), which has less effect on the material (*Formlabs white paper “How Mechanical Properties of Stereolithography 3D Prints are Affected by UV Curing”*). Further, in the mechanical tests of the anisotropic and rectangular materials of both coated and uncoated samples, we did not observe any significant change in the compressive moduli between the samples in either the longitudinal or radial compression. Therefore, modification of the AM parts during the coating process is not a significant concern in this context but may be of consideration for future materials fabrication with surface tension-assisted additive manufacturing.

8. What was a reasoning to select certain AM processes? Why 3 different materials? How not only focusing on one?

We apologize to the reviewer that this point was not originally clear in the manuscript. One objective of this manuscript is to demonstrate that the surface tension-assisted coating is not limited to a specific class of materials or AM process but can be broadly used in laboratories equipped with various 3D printers or AM systems and working with different materials. Therefore, we decided to coat scaffolds made of various materials (*i.e.*, metallic, polymeric), of different geometries, and printed via different AM processes (*i.e.*, powder bed fusion, vat photopolymerization).

To clarify that in the text we have added the following sentence (page 8): *“The method offers a broad range of possibilities in terms of coating materials as well as scaffold shapes and geometries, as long as the surfaces are reticulated, and is not specific to a single additive manufacturing process as demonstrated in Fig 4.”*

9. About the design of the parts: What was the all design parameters used? Now there seems to be only size of the “window” . Thickness of the of parts and pipes? What about the 3D form shown in Figure 1 & 4?

We thank the reviewer for his helpful comment and we agree that design parameters are lacking for some of the 3D parts. To improve clarity and precision, we have added Supplementary Table 1 summarizing the characteristics (*i.e.*, dimensions, window size, pipe diameter, and type of material) of all the different 3D printed scaffolds used in the study.

10. About Mechanical characterization. Amount of samples is quite small. How much variation there was between measurements? What was the repeatability?

Regarding the mechanical characterization of the materials, we used $n = 5$ samples for uncoated scaffolds and $n = 3$ samples for coated scaffolds (as described on page 11) as the variation between measurements was minimal and the measurements were repeatable from sample to sample. The variation between measurements was quantified as the standard deviation, which was plotted with the mean in Figure 5. Further, the individual data points are plotted in this figure. Finally, all raw data plots are shown in the supplementary information for the mechanical characterization (Supplementary Figures 1 and 2).

11. Explain in detail how the experiments of the samples have been planned and analyzed (factors, levels, type of experimental plan, replications, analysis of variance and related statistical tests. Was there predefined experimental design, e.g. a factorial plan? Explain how the process variables (if tested) influence the responses (individual effects and possible interactions).

Throughout the manuscript, the details of how the samples were organized and analyzed has been described. For the data presented in Figures 2, 3, and 6, there was no experimental plan, except for the cell cultures addressed below, as we used these data to demonstrate proof-of-concept and explore the operational space of the principle. In all of these experiments, the presented images are representative of at least $n = 3$ independent experiments of the same condition. For the data in Figure 5, as discussed above, the experiments were planned with two factors (geometry and coating). Each factor had two levels: geometry – rectangular or anisotropic; coating: coated or uncoated. As stated, all measurements for rectangular samples were collected with $n = 5$ and all anisotropic samples were collected with $n = 3$. As this manuscript does not focus on process optimization and we are not arguing at any point in the paper that we have optimized any step of our process, we have not applied design of experiments to isolate precise variation due to the different factors and levels. The conclusion from the mechanical analysis, as stated and as evident from the data, is that design and process influences mechanical properties. A change of two orders of magnitude in compressive modulus in the longitudinal test was observed between rectangular and anisotropic samples whereas the compressive moduli of the rectangular and anisotropic samples were of the same order of magnitude in the radial test. As this method is implemented to design parts for precise applications, a proper design of experiments would be interesting to isolate the process effects of each step of the process, but this is beyond the scope of the current work and is of interest for future pursuit.

12. About the cell cultures, how many samples there were? Parallel and reference samples?

We have added the number of samples for each of the three cell experiments in the manuscript (page 19):

“The experiments were performed on $n=10$ scaffolds for MRC-5 encapsulation, $n=5$ for co-culture, and $n=3$ for high cell density encapsulation.”

These experiments were performed at different days and with parallel samples of cells of different passage numbers.

13. When thinking geometry of the part, how very complex parts with inner structures example pipes could be coated? Will the curing penetrate? With dipping hot to get and make sure that existing resin is get out?

In this work, we have focused on the coating of complex surfaces whereby the surface tension-assisted coating can be leveraged to generate cell-laden, mechanically complex, and multicomponent biomaterials in the shape of a plane, a cylinder, a pyramid, or a polyhedron. In principle, this process can also be used to fill the volume of more complex 3D printed structures but is beyond the scope of this current work. This was not the primary focus here as very thick, three-dimensional constructs require vascularization to remain viable over extended periods of time and the direction of vascularity within our constructs is an active, yet challenging, area of future work. Of note, the coating in this work occurs with low-viscosity liquids and we can dip the

3D-printed scaffolds at ambient temperature or 37 °C, in the case of cell encapsulation, without difficulty of getting the coating material (what the reviewer refers to here as 'resin') into or out of complex structures. Finally, the coating materials are clear solutions and there is negligible attenuation of the light throughout all structures tested and we can, in principle, fabricate structures that are centimeters in thickness without issue.

To clarify this point in the manuscript, we have added the following sentences in Discussion section (page 13):

"Here, we focused on the coating of complex surfaces but this process could also be used to fill the volume of other 3D printed structures."

"In the case of photopolymerization, the coating materials are clear solutions and there is negligible attenuation of the light throughout all structures tested and one should be able to fabricate structures that are micrometers to centimeters in thickness."

14. I had a feeling that too many things (different AM process, materials etc.) has been reported in same paper. I would encourage to deeply think what are the most important findings and limit reporting experiment not related to that. At the moment paper is quite hard to read and takes lot of time to find what sample is related to what test or experiment.

The objective of our manuscript is to describe a general and broad approach that can be used with a range of AM processes, printers, material types and shapes, and polymerization methods. To demonstrate the method versatility, we have deliberately chosen to test a range of conditions. We then focused in more detail on the use of a hydrogel coating of tubular 3D-printed parts generated by vat photopolymerization to generate complex, mechanically anisotropic, and cell-laden biomaterials.

That being said, we understand that the number of variables used in the manuscript might be a bit confusing for the readers and this is why we have tried to clarify the manuscript based on the reviewer's helpful comments and suggestions. More precisely, we have added more detail to the method section: we have added AM process parameters (pages 16-17), and we have clarified which types of coating material were used throughout the study (page 17):

"Versatility of the coating material and polymerization method was demonstrated using four different coating materials (methacrylated gelatin, neat thiol-ene networks, collagen hydrogels, and alginate hydrogels). However, for the rest of the study (i.e., influence of scaffold design parameters, mechanical testing study, and cell culture experiments) methacrylated gelatin coatings were employed."

In addition, we have provided a table (Supplementary Table 1) that summarizes the design characteristics of the 3D parts. We believe that these modifications will improve manuscript clarity for the readers.

I recommend Major revision for the paper to see more about the details and novelty of the submission.

Reviewer #2 (Remarks to the Author):

The paper describes a new method for coating 3D printed reticulated scaffolds with a polymerisable liquid film, through surface tension-driven filling of the fenestrations in the scaffolds.

The phenomenon is explained through modelling the Gibbs free energy with increasing saturation, for a number of different contact angles and ratios of solid-liquid to solid-vapour surface energies.

Examples of coated structures are shown, including a tracheal scaffold that could be optimised for anisotropic mechanical behaviour independent of the coating. The examples also include scaffolds coated with hydrogels encapsulating cells of different types, in cases followed by seeding of a second cell type on the coating surface.

Although the described phenomena must have occurred before in tissue engineering research, to my knowledge this is the first report of it being exploited for the controlled coating and cell seeding of scaffolds.

We are happy to see that the reviewer understood fully the main points and unique aspects of the work. We thank the reviewer for his or her helpful comments, which are addressed specifically below, and we feel that by revising the manuscript accordingly we have produced a more clear and thorough presentation of our findings.

1. It is unclear to me though if it works only for the outer surface, or throughout the bulk of the scaffold (fenestrations within). This is not stated clearly, and some remarks seem contradictory on this point, e.g. 'enables facile encapsulation of mammalian cells throughout the full three-dimensional construct' vs. 'various applications (...) such as hollow biomaterials (...) as well as flat structures'. This should be clarified.

In this manuscript we have focused only on the use of this approach to coat the outer surface of 3D-printed scaffolds. In the demonstrations shown here only the fenestrated mesh on the outer surface of the scaffolds were coated while the inner lumen remains hollow by design. We apologize to the reviewer that this point was unclear in the original version of the manuscript and we agree that some sentences were confusing. Therefore, we have modified the manuscript accordingly (page 3): *"Importantly, the hydrogel coating on the outer surface of the construct enables facile encapsulation of mammalian cells as well as subsequent cell seeding."* In principle, the method can also be applied to filling throughout the bulk of 3D-printed scaffolds and is the subject of on-going work in the lab.

2. The detailed explanation of the phenomenon using a physical chemistry model

strengthens the work. However, the manuscript lacks explanation on how the graphs in Fig. 3 should be interpreted; e.g. is there a max. contact angle for which the technique will work, and if so how can that be seen from the plot?

We thank the reviewer for her/his careful reading of the manuscript and for the appreciation of the physical model of the coating process. We had intentionally left the statement vague in the original manuscript as to the maximum contact angle for which the technique will work, as there are certain assumptions in the model and we have not exhaustively experimented with fluid-scaffold pairs of all contact angles to verify the simple physical model presented here. That said, the model framework suggests that complete coating will not happen beyond a contact angle of $\theta = 90$ degrees. This can be seen in **Fig 3c** as the reduced Gibbs Energy for a saturation of 0 (uncoated scaffold) is lower than that of a saturation of 1 (completely coated scaffold) for all contact angles greater than 90 degrees, implying that past this contact angle an uncoated scaffold is the favorable state. This analysis is in agreement with experimental observations that for hydrophobic materials and aqueous coatings (*i.e.*, contact angles > 90 degrees) complete coating was not observed. To make this clear within the text we have added the following statement (page 7): *“Namely, the reduced Gibbs Energy is lower for $a_L = 0$ than $a_L = 1$ for contact angles greater than 90° (Fig 3c), implying that scaffold-liquid pairs with a large contact angle will prefer the uncoated state.”*

Overall, the analysis suggests that there exists a maximal contact angle for which the technique works; however, this will likely depend in reality on many factors such as the viscosity of the coating fluid; the precise shape, roughness, and dimension of the scaffold struts; and the change in the contact angle and/or volume during the solidification process. A more rigorous fluid mechanical and physical model of the surface tension-driven coating process is a direction of future work and we hope that we can answer this complex aspect of the problem in more detail in the future. For this work, we are quite satisfied that a simple but detailed analysis of the coating phenomenon captures the physical underpinnings of the surface tension-driven coating.

3. A major shortcoming is that despite tissue engineering being named as the main application, none of the used scaffold materials is biodegradable. Why is this only demonstrated for a metal and a number of densely crosslinked, non-resorbable stereolithography resins? It may only be for convenience, but it begs the question if it would work just as well for more relevant materials (*i.e.* biomaterials), which may be 3D printed into scaffolds with lower accuracy, different surface roughness, different contact angles etc.

Ideally it would be demonstrated for more relevant scaffolds; if not it should at least be justified why it would work just as well.

We thank the reviewer for this comment and have now shown that the process is amenable to coating a biodegradable, polycaprolactone scaffold as shown below. We have included this in the Supplementary Information (Supplementary Figure 10). This demonstrates that the process is amenable to more relevant resorbable materials that are commonly employed as biomaterials. We believe this extends the versatility of the concept and motivates how this could easily

transition to use in a biomedical setting.

4. When dip coating a scaffold into a cell suspension, I can imagine there would be a considerable dead volume of suspension that is lost, resulting in a low seeding efficiency. Please elaborate.

We typically use 1 ml of liquid precursor solution to coat the 3 x 1 (l x d) cm tubular devices and roll the scaffolds in the liquid solution to coat it (as shown in **Fig 1b**).

Cylinder outer surface $A = 2\pi r \times l = 2\pi \times 0.5 \times 3 = 9.42 \text{ cm}^2$

Based on imaging, the hydrogel thickness was estimated to be 0.5 mm.

Thus, the volume of the outer shell containing the gel and scaffold $V = 9.42 \times 0.05 = 0.47 \text{ mL}$. As the scaffold comprises ~20% of V , the volume of the gel is ~0.38 mL. In normal practice, we would coat two tubular scaffolds with 1 mL of liquid. Therefore, the dead volume was approximately 0.25 mL or 25%. This has not yet been optimized and we do not believe this is an issue as this is comparable to the dead volume in normal cell encapsulation experiments where an excess of pre-gel solution of a similar order is often prepared.

5. Finally, I don't think the technique should be termed 'Surface tension-assisted additive manufacturing', as the coating is performed after the preparation of a scaffold by additive manufacturing, and the coating procedure does not have the characteristics of additive manufacturing (as are set out in ASTM standards).

The definition of additive manufacturing based on ASTM standards (ASTM/52900) is "Process of joining materials to make parts from 3D model data, usually layer upon layer, as opposed to subtractive manufacturing and formative manufacturing methodologies". Furthermore ASTM specifies that AM processes can be single-step or multi-step processes: "it is rare that a finished

product can be entirely manufactured within a single process principle” and “[the parts] may acquire the geometry in a primary process step and then acquire the fundamental properties of the intended material in a secondary process step.” ASTM gives the following example for metallic parts: “the object acquires the basic geometry by joining material with a binder in the primary process step which is followed by material consolidation by sintering (...) in subsequent process steps.”

In our process, the coating is added to a 3D-printed scaffold in a subsequent step and the process respects the fundamental principle of AM that is “forming three dimensional parts by the successive addition of material”. Importantly, this process remains throughout in direct contrast to subtractive manufacturing and the final part is additively shaped by rational design of the two-stage process. Therefore, given the frame of ASTM standards, we think that this method can be named surface tension-assisted additive manufacturing.

To clarify this point in the manuscript, we have added the following sentences in the Discussion section (page 13):

“The coating is added to a 3D printed scaffold in a subsequent step and the process respects the fundamental principle of additive manufacturing, which is to form three dimensional parts by the successive addition of material (ASTM/52900).”

Reviewer #3 (Remarks to the Author):

This research is well reported and elegant. The dual step technique to address the problem of 3D printing with cells for tissue engineering applications is key. Using the advantages of 3D printing for producing lattice structures and the physical nature of surface tension in these structures to fill in the holes with a secondary material is clever. Applying mechanical design to the lattice hole shapes and sizes to control the mechanical properties and anisotropy of the produced object is a great opportunity to communicate with the wider audiences of Nature. It may allow readers to truly understand that it is not just material properties that control mechanical properties but also the design and process.

We are very pleased to read the positive comments of the reviewer and appreciate her/his careful reading and thorough understanding of the presented work. We fully agree that more emphasis from the community should be placed on how mechanical properties can be directed not only by constituent material properties but also through the design and processing and we hope that this work will contribute to this conversation. To emphasize this point we have added the following sentence to the Discussion (page 14): *“Overall, this work highlights that the mechanical properties of an object are not controlled solely by constituent material properties but also by part design and processing.”*

Reviewers' Comments:

Reviewer #1:

Remarks to the Author:

Overall thanks for good response and explanations to the comments. However it would be necessary to use these explanations to improve the manuscript. Now there seem only minor modifications here and there. Below are listed those comments that should be used to improve the manuscript:

"The Scott reference is not pertinent to the current manuscript as we are using a fundamentally different approach than the UV coatings mentioned in the Scott reference. Here, we employ low-dose UV light to solidify a suspended liquid film in the open windows of a 3D printed scaffold. This is not purely a surface coating in the traditional sense as discussed by Scott, wherein a thin film of a secondary material is deposited uniformly on an initial material, but a unique method to transform a reticulated mesh into a complete three-dimensional and multicomponent object."

This should be opened more and made more clear in this research and manuscript, what is the difference as explained previously.

"The novelty of the research is to leverage surface tension forces to suspend liquid films in the windows of fenestrated structures. Uniquely, the suspended liquid films are transformed into a solid material, often with UV light, and the surface tension coating and solidification is part of constructing the final design. The coated films are not only to modify the surface chemistry of the material but part of the engineered device itself. Although surface tension forces are well described, there is no report to our knowledge of using this physical phenomenon for the controlled fabrication of multicomponent (bio)materials. In the above references mentioned by the reviewer, filling of porous scaffolds was achieved, but the scaffold pore size was small (400 or 500 μm), no evidence for larger window size was demonstrated, and there is no mention of surface tension wetting. In addition, in one of the examples, there is no subsequent polymerization, just an immersion and adsorption of the liquid on the mesh as is common in traditional coating applications. Further, traditional additive manufacturing, especially that in the field of 3D bioprinting, has relied on material deposition or curing at each pixel independently within the final object, thereby coupling the overall time for part fabrication to the volume of the final part. In this work, we uniquely demonstrate that exploiting surface tension-driven coating can enable additive manufacturing of complex, multicomponent materials without the need for pixel-by-pixel fabrication and, here, the speed of the second stage of the process is volume invariant. Thus, the process not only increases the versatility of 3D printing and 3D bioprinting but also presents methods to decrease the speed of fabrication through rational design. Finally, this work demonstrates that the final properties of a 3D-printed part do not rely solely on the constituent material properties but also on the design and fabrication of the part."

This explanation is quite clear and it could be used to improve the manuscript text.

About the AM process parameters:

From metal there is missing: laser powder, scanning speed, exposure pattern track width, hatch distance etc.?

From bioprint orientation and location in the build plate?

"We did not test rigorously how the material properties of the AM parts were changed by photopolymerization of the hydrogel. The dosage used for the photopolymerization of the hydrogel or thiol-ene networks ($I_0 = \sim 6.0 \text{ mW}\cdot\text{cm}^{-2}$; $t = 120 \text{ sec}$) was much lower than that used for the UV post-processing of the 3D-printed devices, which was 30 min. In addition, the photopolymerization of the hydrogel films used a lower wavelength light ($\lambda = 365 \text{ nm}$) compared

to UV curing ($\lambda = 405 \text{ nm}$), which has less effect on the material (Formlabs white paper "How Mechanical Properties of Stereolithography 3D Prints are Affected by UV Curing"). Further, in the mechanical tests of the anisotropic and rectangular materials of both coated and uncoated samples, we did not observe any significant change in the compressive moduli between the samples in either the longitudinal or radial compression. Therefore, modification of the AM parts during the coating process is not a significant concern in this context but may be of consideration for future materials fabrication with surface tension-assisted additive manufacturing."

This should be explained in the manuscript. Main point the different wave length and exposure times.

"We apologize to the reviewer that this point was not originally clear in the manuscript. One objective of this manuscript is to demonstrate that the surface tension-assisted coating is not limited to a specific class of materials or AM process but can be broadly used in laboratories equipped with various 3D printers or AM systems and working with different materials. Therefore, we decided to coat scaffolds made of various materials (i.e., metallic, polymeric), of different geometries, and printed via different AM processes (i.e., powder bed fusion, vat photopolymerization)."

This could be added in the objectives. Proving that it works also with metals and plastics.

"Throughout the manuscript, the details of how the samples were organized and analyzed has been described. For the data presented in Figures 2, 3, and 6, there was no experimental plan, except for the cell cultures addressed below, as we used these data to demonstrate proof-of-concept and explore the operational space of the principle. In all of these experiments, the presented images are representative of at least $n = 3$ independent experiments of the same condition. For the data in Figure 5, as discussed above, the experiments were planned with two factors (geometry and coating). Each factor had two levels: geometry – rectangular or anisotropic; coating: coated or uncoated. As stated, all measurements for rectangular samples were collected with $n = 5$ and all anisotropic samples were collected with $n = 3$. As this manuscript does not focus on process optimization and we are not arguing at any point in the paper that we have optimized any step of our process, we have not applied design of experiments to isolate precise variation due to the different factors and levels. The conclusion from the mechanical analysis, as stated and as evident from the data, is that design and process influences mechanical properties. A change of two orders of magnitude in compressive modulus in the longitudinal test was observed between rectangular and anisotropic samples whereas the compressive moduli of the rectangular and anisotropic samples were of the same order of magnitude in the radial test. As this method is implemented to design parts for precise applications, a proper design of experiments would be interesting to isolate the process effects of each step of the process, but this is beyond the scope of the current work and is of interest for future pursuit."

Again this should be used to improve the text and show also to the readers what has been the plan in the experiments.

Reviewer #2:

Remarks to the Author:

All issues brought up by myself and the other reviewers have been properly addressed, and I would say this is now suitable for publication in Nature Communications.

We are extremely grateful to the reviewers and the editorial team for their time and valuable comments that have helped to improve the revised version of our manuscript. Below, we address these comments individually and summarize the corresponding changes made to the revised version of the original manuscript. Based on the constructive comments of the reviewers, we believe the manuscript is now significantly improved and clearer.

Reviewer #1 (Remarks to the Author):

Overall thanks for good response and explanations to the comments. However it would be necessary to use these explanations to improve the manuscript. Now there seem only minor modifications there and there. Below are listed those comments that should be used to improve the manuscript:

We again thank the reviewer for their helpful input on this manuscript. We have now included additional text to address each of the following points in the revised version of the manuscript as suggested by the reviewer. We believe that this has improved the clarity of the manuscript, especially regarding the innovation and details of the process.

1. "The Scott reference is not pertinent to the current manuscript as we are using a fundamentally different approach than the UV coatings mentioned in the Scott reference. Here, we employ low-dose UV light to solidify a suspended liquid film in the open windows of a 3D printed scaffold. This is not purely a surface coating in the traditional sense as discussed by Scott, wherein a thin film of a secondary material is deposited uniformly on an initial material, but a unique method to transform a reticulated mesh into a complete three-dimensional and multicomponent object."

This should be opened more and make more clear in this research and manuscript, what is the difference as explained previously.

Based on the suggestion of the reviewer and to clarify the coating process used in this manuscript, we have added further details about the process of window generation in the Methods section under 'Surface tension-assisted materials assembly' (page 18) and about the photopolymerization of the suspended liquid films in the Methods section under 'Formation of methacrylated gelatin coatings' (page 18) as well as added additional references for the reader regarding the photopolymerization process as follows:

Surface tension-assisted materials assembly

The procedure for fabricating multicomponent materials via surface tension-assisted additive manufacturing employs the upstream 3D printed mesh scaffold as well as the preparation of the coating precursor solution. The process is similar regardless of the nature of the coating and type of scaffold: the 3D printed scaffold is dipped into a pool of the gel precursor solution, ensuring that all windows are coated uniformly, and crosslinking is induced to transform the suspended liquid film in a solid film. As opposed to a traditional surface coating whereby a thin film of a secondary material is deposited uniformly on a surface, this method transforms the mesh into a three-dimensional and multicomponent object by generating solid windows within the 3D printed scaffold. Versatility of the coating material and polymerization method was demonstrated using four different coating materials

(methacrylated gelatin hydrogels, neat thiol-ene networks, collagen hydrogels, and alginate hydrogels). However, for the rest of the study (*i.e.*, influence of scaffold design parameters, mechanical testing study, and cell culture experiments) methacrylated gelatin coatings were employed.

Formation of methacrylated gelatin coatings

The hydrogel precursor solution was prepared by mixing pre-heated (40 °C) methacrylated gelatin solution (10.0 wt% in UltraPure® distilled water) with LAP (5.0 wt% in UltraPure® distilled water) to final concentrations of 5.0 to 7.5 wt% methacrylated gelatin and 0.5 wt% LAP. The balance of the solutions was composed of UltraPure® distilled water. Photopolymerization was employed to crosslink the methacrylated gelatin solution with an LED-based UV lamp ($\lambda = 365 \text{ nm}$; $I_0 = 6.0 \text{ mW cm}^{-2}$; $t = 120 \text{ s}$; Thorlabs). During irradiation, LAP generates radicals that initiate polymerization of the methacrylate groups of the methacrylated gelatin, ultimately, percolating a network within the suspended liquid film and inducing hydrogel formation constituting a solid window.^{24,26}

24. Loessner, D. *et al.* Functionalization, preparation and use of cell-laden gelatin methacryloyl-based hydrogels as modular tissue culture platforms. *Nature Protocols*. **11**, 727–746 (2016).
26. Fairbanks, B.D. *et al.* Photoinitiated polymerization of PEG-diacrylate with lithium phenyl-2,4,6-trimethylbenzoylphosphinate: polymerization rate and cytocompatibility. *Biomaterials* **30**, 6702–6707 (2009).

Additionally, we have modified the manuscript to clarify the coating process in the Results section under ‘Surface tension-assisted additive manufacturing’ as follows (pages 4-5):

“The suspended liquid films, present in the open windows of the 3D printed scaffolds, were then converted into solid films via an external trigger (temperature, time, light, or ionic gelation) forming a stable solid coating on the reticulated mesh, that covered the previously open windows. The method was validated initially with a methacrylated gelatin (7.5 wt%) coating on a planar mesh scaffold (Somos 9120; DSM). To solidify the suspended liquid films, we induced photopolymerization with low dose UV light [0.5 wt% lithium phenyl-2,4,6-trimethylbenzoylphosphinate (LAP); $\lambda = 365 \text{ nm}$; $I_0 = 6.0 \text{ mW cm}^{-2}$; $t = 120 \text{ s}$] and uniform load-bearing gels were obtained in the windows of the scaffold (**Fig 2a**). In this manner, the approach is not a surface coating in the traditional sense, wherein a thin film of a secondary material is deposited uniformly on an initial material, but a unique method to transform a reticulated mesh into an intact three-dimensional and multicomponent object.”

2. “The novelty of the research is to leverage surface tension forces to suspend liquid films in the windows of fenestrated structures. Uniquely, the suspended liquid films are transformed into a solid material, often with UV light, and the surface tension coating and solidification is part of constructing the final design. The coated films are not only to modify the surface chemistry of the material but part of the engineered device itself. Although surface tension forces are well described, there is no report to our knowledge of using this physical phenomenon for the controlled fabrication of multicomponent (bio)materials. In the above references mentioned by the reviewer, filling of porous scaffolds was achieved, but the scaffold pore size was small (400 or 500 μm), no evidence for larger window size was demonstrated, and there is no mention of surface tension wetting. In addition, in one of the examples, there is no subsequent polymerization, just an immersion and adsorption of the liquid on the mesh as is common in traditional coating applications. Further, traditional additive manufacturing, especially that in the

field of 3D bioprinting, has relied on material deposition or curing at each pixel independently within the final object, thereby coupling the overall time for part fabrication to the volume of the final part. In this work, we uniquely demonstrate that exploiting surface tension-driven coating can enable additive manufacturing of complex, multicomponent materials without the need for pixel-by-pixel fabrication and, here, the speed of the second stage of the process is volume invariant. Thus, the process not only increases the versatility of 3D printing and 3D bioprinting but also presents methods to decrease the speed of fabrication through rational design. Finally, this work demonstrates that the final properties of a 3D-printed part do not rely solely on the constituent material properties but also on the design and fabrication of the part.”

This explanation is quite clear in it could be used to improve the manuscript text.

Based on the suggestion of the reviewer and to clarify the innovation of this work, we have clarified the text in the Discussion section as follows (page 14):

“The main innovation of this approach is to leverage surface tension to suspend stable liquid films across the fenestrations of a lattice after immersion in a hydrogel precursor solution. The liquid film is then crosslinked to generate solid windows that bear weight, hold pressure, and, in some cases, comprise a tunable niche for cell seeding and encapsulation. Although surface tension forces are well described, there is no report to our knowledge that exploits this physical phenomenon for the controlled fabrication of multicomponent (bio)materials in this manner. Importantly, the solid films not only modify the surface chemistry of the material but constitute an integral part of the engineered device itself. Traditional additive manufacturing, especially in the field of 3D bioprinting, has relied on material deposition or curing at each pixel independently within the final object, thereby coupling the overall time for part fabrication to the volume of the final part. In this work, we demonstrated that surface tension-driven coating can enable additive manufacturing of complex materials without the need for pixel-by-pixel fabrication and, here, the speed of the second stage of the process was volume invariant across the range of devices tested. Thus, the process not only increases the versatility of 3D bioprinting but also decreases the speed of fabrication through rational design. Further, this approach highlights that the final properties of a 3D printed part do not rely solely on the material properties of the constituent materials but also on the design and fabrication of the part. Bio-integration, elasticity, lightness, and an intact seal can be imparted by the hydrogel coating while structural integrity and anisotropic mechanical properties are conferred by the 3D printed backbone. Here, we focused on the coating of complex surfaces but this process could also be used, in principle, to fill the volume of other 3D printed structures.”

3. About the AM process parameters:

From metal there is missing: laser powder, scanning speed, exposure pattern track width, hatch distance etc.?

From bot print orientation and location in the build plate?

We have added the laser power and the print orientation in the Methods section under ‘Additive manufacturing of reticulated scaffolds’ (page 17). However, the other parameters were not available from the 3D printing company used for the fabrication of the metal parts (Proto Labs, Maple Plain, MN, USA).

“The metal pyramids were fabricated by powder bed fusion (laser power 90 W) using a ConceptLaser ML2 DMLS printer (Proto Labs, Maple Plain, MN, USA) in stainless steel 17-4 PH (fineline™, Rayleigh,

NC, USA). The print orientation was such that the base of the pyramid was built on the build plate with the apex directly above the centroid of the base.”

4. “We did not test rigorously how the material properties of the AM parts were changed by photopolymerization of the hydrogel. The dosage used for the photopolymerization of the hydrogel or thiol-ene networks ($I_0 = \sim 6.0 \text{ mW.cm}^{-2}$; $t = 120 \text{ sec}$) was much lower than that used for the UV post-processing of the 3D-printed devices, which was 30 min. In addition, the photopolymerization of the hydrogel films used a lower wavelength light ($\lambda = 365 \text{ nm}$) compared to UV curing ($\lambda = 405 \text{ nm}$), which has less effect on the material (Formlabs white paper “How Mechanical Properties of Stereolithography 3D Prints are Affected by UV Curing”). Further, in the mechanical tests of the anisotropic and rectangular materials of both coated and uncoated samples, we did not observe any significant change in the compressive moduli between the samples in either the longitudinal or radial compression. Therefore, modification of the AM parts during the coating process is not a significant concern in this context but may be of consideration for future materials fabrication with surface tension-assisted additive manufacturing.”

This should be explained in the manuscript. Main point the different wave length and exposure times

To address the original comment of how the photopolymerization of the coating may affect the material properties of the 3D printed structure, the following text was added to the Discussion section (pages 14-15):

“The total dosage of UV light for the photopolymerization ($\lambda = 365 \text{ nm}$; $t = 120 \text{ s}$) was much lower than that for the post-processing of the 3D printed devices ($\lambda = 405 \text{ nm}$; 30 min). Further, we did not observe any significant change in the compressive moduli between the uncoated and coated devices. Therefore, modification of the mechanical properties of the parts during the photopolymerization of the coating was not a significant concern in this context but should be considered in future material fabrication.”

5. “We apologize to the reviewer that this point was not originally clear in the manuscript. One objective of this manuscript is to demonstrate that the surface tension-assisted coating is not limited to a specific class of materials or AM process but can be broadly used in laboratories equipped with various 3D printers or AM systems and working with different materials. Therefore, we decided to coat scaffolds made of various materials (i.e., metallic, polymeric), of different geometries, and printed via different AM processes (i.e., powder bed fusion, vat photopolymerization).”

This could be added in the objectives. Proving that it works also with metals and plastics.

As suggested by the reviewer we have modified the manuscript to emphasize the broad utility of this method in two places. First, in the Introduction (page 3) as follows:

“Furthermore, the process is versatile in material design and does not rely on any specialized equipment; the method is agnostic with respect to the upstream 3D printer, scaffolding material (e.g., resins, metals, biodegradable polymers), and class of hydrogel.”

Second, in the Results section under ‘Material versatility and geometric control’ (page 8) as follows:

“Surface tension-assisted additive manufacturing is not limited to a specific class of materials or additive manufacturing processes but can be used broadly in laboratories equipped with various 3D printers and working with different materials. Here, we have utilized scaffolds made of various materials (i.e., metal, polymer), of different geometries, and printed with different additive manufacturing processes (i.e., powder bed fusion, vat photopolymerization) as demonstrated in **Fig 4** and **Supplementary Figure 10.**”

6. “Throughout the manuscript, the details of how the samples were organized and analyzed has been described. For the data presented in Figures 2, 3, and 6, there was no experimental plan, except for the cell cultures addressed below, as we used these data to demonstrate proof-of-concept

and explore the operational space of the principle. In all of these experiments, the presented images are representative of at least $n = 3$ independent experiments of the same condition. For the data in Figure 5, as discussed above, the experiments were planned with two factors (geometry and coating). Each factor had two levels: geometry – rectangular or anisotropic; coating: coated or uncoated. As stated, all measurements for rectangular samples were collected with $n = 5$ and all anisotropic samples were collected with $n = 3$. As this manuscript does not focus on process optimization and we are not arguing at any point in the paper that we have optimized any step of our process, we have not applied design of experiments to isolate precise variation due to the different factors and levels. The conclusion from the mechanical analysis, as stated and as evident from the data, is that design and process influences mechanical properties. A change of two orders of magnitude in compressive modulus in the longitudinal test was observed between rectangular and anisotropic samples whereas the compressive moduli of the rectangular and anisotropic samples were of the same order of magnitude in the radial test. As this method is implemented to design parts for precise applications, a proper design of experiments would be interesting to isolate the process effects of each step of the process, but this is beyond the scope of the current work and is of interest for future pursuit.

Again this should be used to improve the text and show also to the readers what has been the plan in the experiments.

As suggested by the reviewer and to more clearly articulate the manner in which the experiments were planned and analyzed, we have modified the manuscript in multiple places. First, in the Methods under ‘Cell-laden materials and confocal imaging’ (page 20) as follows:

“The experiments were performed on n=10 scaffolds for MRC-5 encapsulation, n=5 for co-culture, and n=3 for high cell density encapsulation.”

Second, in the Methods section under ‘Mechanical testing’ (page 19) as follows:

“The experiments were planned with two factors: geometry and coating. Each factor had two levels – geometry: rectangular or anisotropic; coating: uncoated or coated. Measurements of the rectangular devices (uncoated and coated) were performed on n=5 samples and measurements of the anisotropic devices (uncoated and coated) were performed on n=3 samples.”

Third, in the Discussion section (page 10) as follows:

“In these studies, design of experiments was not applied to isolate precise variation in mechanical properties caused by different processing conditions. Yet, it is clear that design and process influence the final mechanical properties of the device. A thorough investigation of the effects on each stage of the fabrication process on final part properties would be of interest but is beyond the scope of the current work.”

Reviewer #2 (Remarks to the Author):

All issues brought up by myself and the other reviewers have been properly addressed, and I would say this is now suitable for publication in Nature Communications.

We thank the reviewer for their time and helpful comments.